



# Consistent CryoSat-2 and Envisat Freeboard Retrieval of Arctic and Antarctic Sea Ice

Stephan Paul[1], Stefan Hendricks[1], Robert Ricker[1], Stefan Kern[2], and Eero Rinne[3]

[1]Alfred Wegener Institute, Helmholtz Centre for Polar and Marine Research, Bremerhaven, Germany
[2]Integrated Climate Data Center, Hamburg, Germany
[3]Finnish Meteorological Institute, Helsinki, Finland

*Correspondence to:* Stephan Paul (stephan.paul@awi.de)

**Abstract.** In order to derive long-term changes in sea-ice volume, a multi-decadal sea-ice thickness record is required. CryoSat-2 has showcased the potential of radar altimetry for sea-ice mass-balance estimation over the last years. However, precursor altimetry missions such as Envisat have not been exploited to the same extent so far. Combining both missions to acquire a decadal sea-ice volume data set requires a method to overcome the discrepancies due to different foot-print sizes from either pulse-limited or beam-sharpened radar echoes. In this study, we implemented an inter-mission consistent surface-type classification scheme for both hemispheres, based on the waveform pulse peakiness, leading-edge width, and sea-ice backscatter. In order to achieve a consistent retracking procedure, we adapted the Threshold First Maximum Retracker Algorithm, previously used only for CryoSat-2, to develop an adaptive retracker threshold that depends on waveform characteristics. With our method, we produce a global and consistent freeboard data set for CryoSat-2 and Envisat. This novel data set features a maximum monthly difference in the mission-overlap period of 2.2 cm (2.7 cm) for the Arctic (Antarctic) based on all gridded values with spatial resolution of 25 km × 25 km and 50 km × 50 km for the Arctic and Antarctic, respectively.

## 1 Introduction

The Arctic sea-ice cover has reduced over the last decades (e.g., Stroeve et al., 2012; Meier et al., 2014). The Antarctic sea-ice cover has been slightly increasing (Parkinson and Cavalieri, 2012; Parkinson and DiGirolamo, 2016) but is subject to substantial inter-annual variation (e.g. Turner et al., 2017). Aside from the reduction in extent, the Arctic sea ice is also thinning. This thinning was observed by a variety of sensors such as upward-looking sonar measurements from submarines, aircraft measurements, as well as autonomous measurements (e.g., Rothrock et al., 1999; Meier et al., 2014; Lindsay and Schweiger, 2015). For the Antarctic, however, our knowledge about changes in the sea-ice thickness is much more limited than for the Arctic. Only few localized measurements are available from upward-looking sonars (e.g., Behrendt et al., 2013), drillings (e.g., Ozsoy-Cicek et al., 2013), as well as ship- and airborne measurements (e.g., Haas, 1998; Haas et al., 2008; Leuschen et al., 2008; Worby et al., 2008a). A different approach is the use of satellite laser altimetry utilizing the Ice, Cloud and land Elevation Satellite (ICESat; e.g., Farrell et al., 2009; Kwok and Rothrock, 2009). While this approach benefits from a very small sensor footprint, ICESat data is limited temporarily to autumn and spring acquisition seasons as well as spatially through present cloud cover. It is widely accepted however that complete coverage of both polar regions can be achieved only





with satellite-altimetry (Laxon et al., 2003; Giles et al., 2007; Kwok et al., 2009; Kurtz and Markus, 2012; Laxon et al., 2013; Kern et al., 2016; Schwegmann et al., 2016; Ricker et al., 2017; Tilling et al., 2017).

The general methodology of retrieving sea-ice freeboard and sea-ice thickness using satellite radar altimetry is based on the pioneering work of Laxon et al. (2003) as well as Peacock and Laxon (2004). In a first step, the echo power waveforms

from the quasi-nadir run-time measurement are classified as returns from either sea-ice floes or returns from the sea surface of leads between sea-ice floes. These measurements are then converted into distance measurements which are so accurate that one can see the difference in elevation of the snow surface or the sea-ice surface relative to the sea surface in the leads. Here, one can differentiate between the total freeboard (i.e., the height difference between the top of the snow surface and the sea surface) and the sea-ice freeboard (i.e., the height difference between the sea-ice surface and the sea surface). However, the

assumption that the retrieved distance using Ku-band radar always coincides with the snow/ice interface is not true, especially for a highly stratified sea-ice snow cover and/or for multi-year sea-ice regimes (e.g., Armitage and Ridout, 2015). Therefore, the retrieved freeboard from the altimeter is often referred to as radar freeboard. Additionally, a correction for the lower wave propagation speed in the sea-ice snow cover needs to be applied. The total sea-ice thickness can then be calculated from the sea-ice freeboard by assuming hydrostatic equilibrium (e.g., Ricker et al., 2014).

The objective of the European Space Agency's (ESA) Climate Change Initiative for sea ice (SICCI) is to achieve a consistent sea-ice freeboard and sea-ice thickness climate data record (CDR) for both polar regions by combining radar altimetry data from all available missions with full error characterization. For our study we use data from the Environmental Satellite (Envisat) as well as CryoSat-2. Envisat carries a the pulse-limited Radar Altimeter (RA) whereas CryoSat-2 on the other hand utilizes the along-track beam-sharpened Synthetic Aperture Interferometric Radar Altimeter (SIRAL).

During the first phase of SICCI (SICCI-1), the focus was set on creating a processing scheme for Envisat data with the possibility to derive Arctic and Antarctic sea-ice freeboard and sea-ice thickness (Schwegmann et al., 2016). Here, the surface-type classification was based solely on the use of a single classifier to positively identify waveforms as either sea ice or leads from otherwise mixed waveform records. In general, this resulted in very few classified sea-ice-type waveforms and in turn comparably high lead fractions for the Antarctic (Schwegmann et al., 2016), but also for the Arctic. As a consequence of the

very low amount of sea-ice-type classifications, only a very coarse resolution of $100\,\mathrm{km} \times 100\,\mathrm{km}$ could be realized for the gridded final data product due to otherwise insufficient coverage.

Furthermore, two different retracking schemes for Envisat were employed for lead-type and sea-ice-type waveforms. For lead-type waveforms, a retracker based on multiple fitting functions was used (Giles et al., 2007), whereas sea-ice-type waveforms were retracked by utilizing the standard offset-center-of-gravity (OCOG) retracker (Wingham et al., 1986). On the other

side, studies such as Ricker et al. (2014) utilize multi-parameter threshold approaches for CryoSat-2 data to differentiate between lead-type and sea-ice-type waveforms and employ a threshold first maximum retracker algorithm (TFMRA; Helm et al., 2014; Ricker et al., 2014) to both. Inconsistencies were also present in the use of differing auxiliary data sets for sea-ice concentration, as well as snow and sea-ice type information. Additionally, different sensor configurations result in varying instrument footprints with associated discrepancies in the degree of surface-type mixing.





In this study, we focus on deriving an inter-mission consistent waveform interpretation scheme over sea-ice areas for Envisat and CryoSat-2 in the framework of the second phase of SICCI (SICCI-2). The main challenge is to build a consistent sea-ice freeboard data record that takes into account the different sensor configurations and differing footprints between both sensors. Therefore, we have developed an empirical approach to minimize inter-mission biases in the surface-type classification as well

as in the range retracking and subsequent freeboard retrieval based on CryoSat-2 reference data for the mission-overlap period (MOP) from November 2010 to March 2012. The resulting parametrization takes into account differences between sea-ice surface properties in both hemisphere as well as the seasonal cycle. In this study we focus on the derivation of freeboard since the conversion from freeboard to thickness is identical for both missions and relies on additional auxiliary data sets.

In the following sections we describe the derivation of a mutual threshold-based surface-type classification from a mix of

unsupervised clustering and supervised classification. Additionally, the derivation and application of a waveform-parameter dependent adaptive threshold retracker scheme for the Envisat freeboard retrieval is presented. Resulting data sets and key benchmarks from Envisat and CryoSat-2 for the MOP are then presented and discussed.

## 2 Data and Methods

This section gives an overview about the used input data and necessary pre-processing and filtering steps. Moreover, we

describe the inter-mission consistent surface-type classification and range retracking scheme.

### 2.1 Input data

#### 2.1.1 Altimetry data

For our study, we use geolocated level 1b (L1b) data for both CryoSat-2 and Envisat. In case of CryoSat-2, we make use of all available SIRAL Baseline-C data acquired in synthetic aperture radar mode (SAR) as well as in the SAR interferometric (SIN)

mode. However, the specific interferometric information is not used during the processing. For Envisat, we use version 2.1 of the sensor geophysical data record (SGDR). All data is provided by the ESA.

#### 2.1.2 Auxiliary data

For our surface-type classification, as well as for the conversion of elevations to sea-ice freeboard, we utilize a range of different auxiliary data sets. Our objective is to consequently maintain methodological as well as auxiliary data consistency.

This is especially important for a multi-mission climate data record.

In this study, for both hemispheres we use the sea-ice concentration data obtained from the Ocean and Sea Ice Satellite Application Facility (OSISAF) as well as the mean sea-surface height product provided by the Danish Technical University (DTU) in its 2015 version. Sea-ice concentration data is used mainly for waveform filtering, whereas the mean sea-surface height data is utilized to eliminate undulations due to the geoid before retrieving sea-ice freeboard (Ricker et al., 2014). We

use the same sea-ice concentration and mean sea-surface height data for both hemispheres.



Additionally, information about the sea-ice snow cover are required. These are necessary for the range correction due to the lower wave-propagation speed in the snow pack. For the Arctic, we use the Warren snow climatology (Warren et al., 1999). As the Warren climatology is based on data sets obtained from Arctic drift stations primarily on multi-year sea ice (MYI), snow-depth values are suspected to be biased high in first-year sea-ice (FYI) regime. Therefore, we apply a correction to the Warren

climatology over FYI (Kurtz and Farrell, 2011). As a consequence, the correction is a linear proportional reduction of the original snow depth with the present FYI fraction down to 50 % of its original value over sole FYI. In order to discriminate between FYI and MYI in the Arctic, we use a MYI fraction data set based on the Special Sensor Microwave Imager (SSM/I)/Special Sensor Microwave Imager Sounder (SSMIS) sensors on-board of the Defense Meteorological Satellite Program (DMSP) satellites provided by the Integrated Climate Data Center (ICDC). This MYI fraction data set is tailored to be consistent for the

entire ERS 1/2 - Envisat - CryoSat-2 period. It is based on NSIDC daily gridded 25 km grid resolution brightness temperatures (Maslanik and Stroeve, 2004, updated 2017) of DMSP-f11, DMSP-f13 and DMSP-f17, inter-sensor calibrated to the level of DMSP-f17 SSMIS measurements. The MYI fraction is computed using the NASA-Team algorithm (Cavalieri et al., 1999) with monthly MYI and FYI tie points computed from inter-sensor calibrated brightness temperatures using a gradient-ratio (at 37 and 19 GHz vertical polarization) threshold approach following an idea formulated in Comiso (2012); monthly open water

tie points are computed from the same data set from grid cells with the 2 % lowest brightness temperatures over open water. The resulting MYI area computed from the obtained MYI fraction data set agrees well with the results of Comiso (2012) and Kwok and Cunningham (2015). More information is given in Kern (unpublished manuscript, 2016).

For the Antarctic, we assume only a single sea-ice type being present. As the Warren climatology is only available for the Arctic, we use a snow-depth climatology derived from data acquired by the Advanced Microwave Scanning Radiometer-EOS

(AMSR-E) and AMSR-2 aboard GCOM-W1 for the Antarctic (Kern et al., 2015; data access via: http://icdc.cen.unihamburg.de/ projekte/esa-cci-sea-ice-ecv0.html). This data set is based on a revised version of the approach described by Markus and Cavalieri (1998) and Markus et al. (2011). Daily snow depths of 13 full seasonal cycles (August through July of years 2002/03 through 2010/11 and years 2012/13 through 2015/16) are used to compute a daily Antarctic snow depth on sea ice climatology. Note that even though this climatology is based on snow depth data derived with a version of the original empirical algorithm

which is now developed directly from AMSR-E brightness temperatures (Frost et al., 2015), the limitations of the algorithm in terms of snow depth under-estimation over deformed sea ice and snow depth sensitivity to snow properties such as wetness (e.g., Worby et al., 2008b; Kern and Ozsoy-Çiçek, 2016) essentially remain the same.

## 2.2 Pre-processing and filtering

As a first step, general filtering is applied. For Envisat this means investigating the measurement-confidence data flags in the

SGDR for problematic records. All data with 'Packet Length Error' (Flag 0), invalid OnBoard Data Handling (Flag 1), an Automatic Gain Control fault (Flag 4), a Rx Delay Fault (Flag 5) or an Waveform Fault (Flag 6) raised are removed from processing. For the CryoSat-2, no additional data filtering is conducted.

In the second step, all input data is filtered regionally by latitudinal boundaries to areas where sea ice is present. This is done for both hemispheres. Data are only considered if located north of 60 °N for the Arctic and south of 50 °S for the Antarctic.

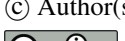

Finally, all processing for both sensors is limited to waveforms flagged as ocean.

## 2.3 Surface-type classification

### 2.3.1 Importance and general issues

The surface-type classification is a crucial part in the processing chain, because the detection of leads is essential for deter-
mining the sea-surface height as present day sea-surface height products are not reliable and accurate enough to be used in
the polar regions right away. The sea-surface height in turn is used as the reference from which the sea-ice freeboard is calcu-
lated. Moreover, a clear distinction between leads and sea ice improves the quality and accuracy of resulting sea-ice freeboard
estimates. Ambiguous signals are excluded from the freeboard retrieval.

In general, with smaller instrument footprint sizes, less surface-type mixing occurs. However, leads often dominate acquired
waveforms due to their specular reflection, and therefore represent sources of strong backscatter. If leads are located off nadir
then their strong backscatter can substantially decrease the quality of the range retracking. In case of Envisat, the nominal
circular footprint of $2\,\mathrm{km}$ in diameter (Connor et al., 2009) can increase to up to $10\,\mathrm{km}$ (Chelton et al., 1989) for strong off-
nadir backscatter sources. Despite its much smaller footprint ($1.65\,\mathrm{km}\times0.30\,\mathrm{km}$), CryoSat-2 can also be affected by off-nadir
leads, which will result in erroneous freeboard estimates (Armitage and Davidson, 2014).

In contrast to the work conducted during SICCI-1, where a single threshold classification scheme for Envisat was used along-
side a multi-parameter classification scheme for CryoSat-2, we aim for a inter-mission consistent surface-type classification
scheme for Envisat and CryoSat-2. Therefore, a set of classifiers that is available for both sensors is necessary. Here, we use
the sea-ice backscatter, the leading-edge width, and the pulse peakiness as classifiers to identify lead-type and sea-ice-type
waveforms from mixed- or ambiguous-type waveforms.

We define pulse peakiness ($pp$) slightly different as compared to the one used by Laxon et al. (2003). Ours follows the
definition of Ricker et al. (2014), where $N_{wf}$ is the number of range bins, $wf_i$ is the echo power at range bin $i$ of the waveform,
and $max(wf)$ is the maximum echo power in the given waveform:

$$pp = \sum_{i=1}^{N_{wf}} \frac{max(wf)}{wf_i} \times N_{wf} \tag{1}$$

The leading-edge width is defined as the width in range bins along the power rise to the first local maximum between $5\,\%$
and $95\,\%$ of the first-maximum peak power while using a ten-times oversampled waveform.

The choice for using three classifiers in SICCI-2 also allows for less strict thresholds compared to the previously used single
threshold parameter classification for Envisat during SICCI-1.

Over the course of a winter season, ice conditions can change substantially. Similar to leads, young- and thin-ice areas
cause specular reflections compared to other ice types. Furthermore, the amount of leads varies both seasonally and regionally.
Based on fixed thresholds for a whole winter season, these changes are difficult to capture and the rejection rate is increased
unnecessarily. Hence, we decided on using monthly thresholds to improve the overall results and data quality.





There is a general lack of ground-truth data as collocated measurements of the same sea-ice situation are very difficult due to sea-ice drift. However, received waveforms have very distinct characteristics and are well described in literature (e.g., Ricker et al., 2014; Schwegmann et al., 2016). These characteristics can also be deduced from the chosen set of classifiers. In order to overcome the lack of ground-truth, we decided to use a combination of unsupervised clustering and supervised classification.

Based on this combination, we are able to determine suitable thresholds for data acquired by Envisat as well as CryoSat-2. The work-flow of how we derived the surface-type thresholds is summarized in Figure 1 and described thoroughly in the following subsections.

### 2.3.2 Monthly classifiers and k-means clustering

In a first step, the three classifiers are computed for all available L1b data per sensor and month in the mission overlap period
(MOP) from November 2010 to March 2012. We only use waveforms that are located between $70\,°N$ and $81.5\,°N$ for the Arctic and feature a minimum sea-ice concentration of $70\,\%$. The northern limit of 81.5°N was chosen to assure a maximum of consistency between Envisat and CryoSat-2 with their differing orbital parameters. Until an update to the geographic mode mask in July 2014, CryoSat-2 operated in SIN mode in an area between 80–85 °N and 100–140 °W (referred to as "Wingham Box"). For this area, as well as all other Arctic areas that are covered while CryoSat-2 operates in SIN mode, we use all
waveforms acquired north of $70\,°N$. For the Antarctic the same restrictions apply, but waveforms are geographically limited to an area south of 65 °S to exclude the majority of the marginal-ice zone to reduce the impact of ocean swell.

Next, a subset of $1\,\%$ is sampled at random without replacement (i.e., each original data point can only appear once) for each month in the MOP from each computed monthly classifier set and for each sensor independently. This data sample is then separated into three clusters using k-means clustering (MacQueen, 1967; Hartigan and Wong, 1979). This methodology
is widely used to separate input data of $N$ observations into $K$ clusters of equal variance, whereby the within-cluster sum-of-squares are minimized (MacQueen, 1967; Hartigan and Wong, 1979).

Generally, the preselection of the number of clusters can be a problem when utilizing k-means clustering. However, while we tested a higher number of initial clusters with perspective of later reunion of similar clusters, a separation into just three clusters turned out to be sufficient. Overall, lead waveforms account for a smaller fraction of the total measurements than
sea-ice waveforms. Because of this and the fact that k-means clustering tends towards generating equal-size clusters (this is a presumption of k-mean algorithms), sole use of k-means clustering for the complete data set was not feasible.

### 2.3.3 Random forest classification

As k-means clustering can not be used for classification of the complete data set due to its unevenly distributed nature, the initially clustered $1\,\%$ data sample is instead used as a-priori information to train a random forest (Breiman, 2001). Random
forests are an ensemble supervised machine-learning method used for classification, among other tasks. This classification is based on a large number of single decision trees that are fitted to randomized sub-samples of the given training data set (Breiman, 2001). After initial training, the random forest can be used for classification of the remaining data. Each decision tree in the trained forest then does a classification and casts a unique vote. In the end, the majority decides the resulting class.





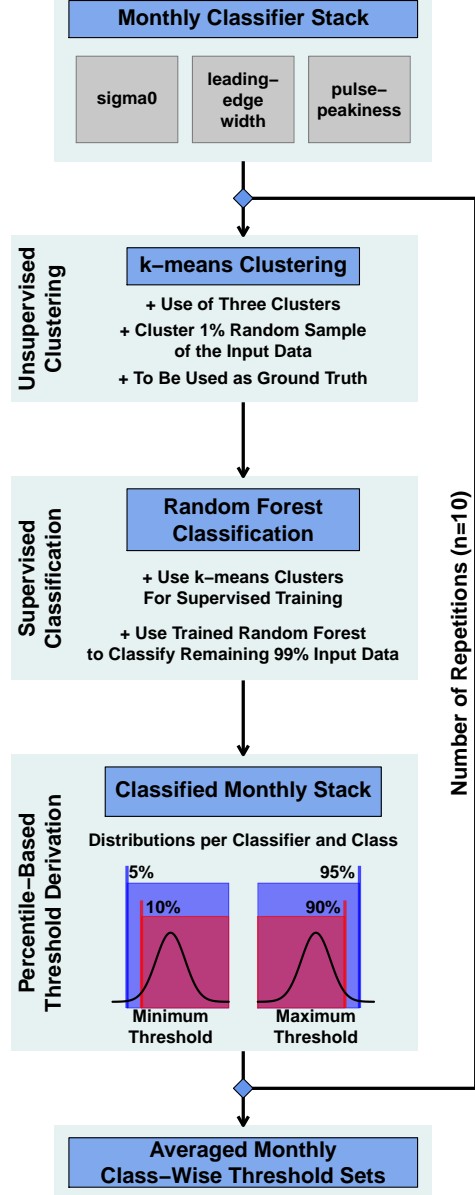

**Figure 1.** Flowchart visualizing the important sub steps of unsupervised clustering and supervised classification in order to derive the new surface-type thresholds from monthly stacks of sea-ice backscatter, leading-edge width and pulse peakiness.

In a random forest, each decision tree is grown following certain rules: First, from the training data of size $N$, $N$ cases are sampled randomly with replacement as a specific training data set for each single tree. This means that the resulting training data set for each tree has the same size as the input data, but any single unique data point can appear multiple times (i.e., "with





replacement"). Second, for $M$ input parameters (in our case sea-ice backscatter, pulse peakiness, and leading-edge width), Breiman (2001) states that ideally a fixed number $m \ll M$ of the given input parameters is specified and randomly selected out of $M$. The best split on these selected parameters $m$ is then used to split the node. Throughout the growing of the forest, the value of $m$ is held constant. Third, each tree is grown out fully, i.e., to its largest possible extent. No pruning is applied.

In contrast to single decision trees that tend to overfit, random forests do not overfit and are also capable of dealing with unbalanced data sets (Breiman, 2001). For our purpose, we always grow a total number of 500 decision trees per training. Due to the small number of input parameters ($M = 3$), we set $m$ to one, following the suggestion by Breiman (2001) to approximate $m$ by $\sqrt{M}$.

Available data from months that are covered twice during the mission-overlap period are merged for the random-forest
training. The trained random forest for each month is then used to classify the remaining 99 % of the corresponding monthly data. From this classified data set, distributions for each of the three classifiers for each month in the mission-overlap period are obtained. These distributions feature clear distinctions for each surface-type class (leads, sea ice, and ambiguous). For example, leads feature in general high values in sea-ice backscatter and pulse peakiness as well as shorter leading-edge widths. The opposite can be seen for sea ice. The class of ambiguous signals is placed in between.

### 2.3.4  Percentile-based averaged thresholds

Thresholds are then obtained from the resulting classifier distributions by using either the 5 % or 10 % percentile for a minimum threshold, or the 90 % or 95 % percentile in case of a maximum threshold (Figure 1). The exact numbers were chosen arbitrarily after visual screening of all resulting classifier distributions to eliminate outliers. The choice of using the more strict (10 %/90 %) or less strict (5 %/95 %) percentile thresholds depends on the sensor. Due to its larger footprint and therefore an
expected higher degree of surface-type mixing, we chose the more strict thresholds for Envisat, and the less strict thresholds for CryoSat-2 due to its smaller footprint.

The decision, whether to derive a minimum or maximum threshold depends on the surface-type class. For example, lead-type waveforms are generally characterized high values in pulse-peakiness as well as sea-ice backscatter due to their specular reflection. Lead-type waveforms therefore feature a very steep increase in echo power which results in short leading-edge
widths. Hence, the 5 %/10 % percentiles of the sea-ice backscatter and pulse-peakiness distributions would be used alongside the 90 %/95 % percentile of the leading-edge-width distribution. Sea-ice type waveforms on the other hand should have smaller values in pulse peakiness. Due to their rather diffuse reflection, sea-ice type waveforms also feature low backscatter and a less steep increase in echo power, which results in longer leading-edge widths. As a result, the 90 %/95 % percentiles are used for the sea-ice backscatter and pulse-peakiness distributions. For the leading-edge width, the 5 %/10 % percentile of its distribution
is used. As we positively identify sea-ice and lead waveforms from all available measurements, all remaining waveforms are classified as ambiguous.

Additionally, for all classifications of leads as well as sea ice in both hemispheres we set a minimum requirement of 70 % sea-ice concentration.



The whole procedure, starting with randomly sampling 1 % from the initial monthly stack, is then repeated ten times. As the whole procedure is initially based on random sampling, this repetition is done to compensate for the odd case of an insufficient representation of lead or sea-ice waveforms in the sampled data. In a last step, the average minimum/maximum thresholds for each classifier, surface-type class, and month in the MOP are estimated for each sensor. These thresholds are summarized in
Tables A1 through A6 in the appendix.

## 2.4   Range retracking and freeboard retrieval

The range-retracking algorithm for Envisat and CryoSat-2 waveforms is identical for sea-ice-type and lead-type waveforms. The used Threshold First Maximum Retracker Algorithm (TFMRA, Helm et al., 2014; Ricker et al., 2014) is based on the following steps:

– Either estimating the noise level as the average of the first five bins of the waveform (CryoSat-2) or discarding all counts in the first five bins of the waveform as these just contain artifacts of the fast Fourier transformation (Envisat);

– Oversampling of the waveforms by a factor of 10 using linear interpolation;

– Smoothing of the oversampled waveforms with a window filter size of 11 range bins;

– Locating the first local maximum of the waveform. This maximum has to be higher than the noise level by 15 % of the
absolute peak power; and finally,

– Obtaining the range value (i.e., the elevation) at a specified percentage threshold of the power at the detected first maximum, by linear interpolation of the smoothed and oversampled waveform.

The conversion from range estimates into sea-ice freeboard follows by subtracting the interpolated sea-surface height (the sum of mean sea-surface height taken from the DTU2015 product and the instantaneous sea-surface height anomaly estimated
from the interpolated elevation between present leads) at the floe location from the elevation of the sea-ice floe. Given a wave-propagation speed correction based on the auxiliary snow depth data, sea-ice freeboard can be calculated. A thorough description on the calculation of sea-ice freeboard (and also sea-ice thickness which is not part of this study) is described in Ricker et al. (2014).

Continuing on the last point of the general TFMRA retracking procedure, the choice of retracker threshold is pivotal for the
range estimation. Following the AWI's implementation for CryoSat-2 (Ricker et al., 2014), we keep a consistent threshold of 50 % from the first maximum peak power both for lead-type and sea-ice-type waveforms. For pulse-limited altimetry such as Envisat, retracking near the maximum power for leads proved to be essential to retrieve reasonable sea-ice freeboard estimates (e.g., Giles et al., 2007). Hence, we chose a threshold of 95 % for leads from Envisat waveforms. In a very recent study by Guerreiro et al. (2017), the use of a 50 % for lead-type wave forms resulted in initial average conditions where the lead surface
elevation was detected above that of the surrounding sea-ice floes. However, when we used a single fixed threshold for the range retrieval over sea ice similar to that for CryoSat-2 (i.e., a 50 % threshold from the first local maximum peak power), our





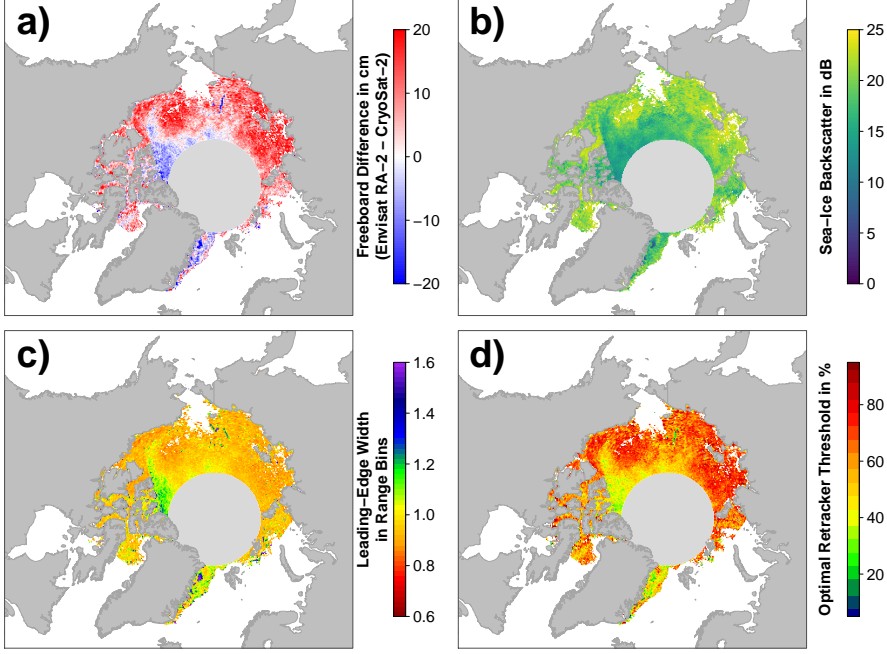

**Figure 2.** Exemplary visualizations of freeboard differences between Envisat and CryoSat-2 (a; in cm), Envisat sea-ice backscatter (b; in dB), Envisat leading-edge width (c; in range bins), and optimal retracker threshold (d; in %) for the Arctic in November 2011.

Envisat sea-ice-freeboard estimates featured an overall smaller variation and range than CryoSat-2 estimates. We relate this behavior to the much larger footprint and the therefore increased mixing of surface types of different surface-roughness scales in every obtained Envisat waveform.

We used our methodology as illustrated in the previous paragraphs and in Figure 1 to compute the sea-ice freeboard for every month of the MOP separately for Envisat and CryoSat-2. Subsequently, we computed the sea-ice freeboard difference Envisat minus CrysoSat-2 , which is shown together with parameters of the retrieval in Figures 2 and 3. From Figures 2a and 3a it appears that there are substantial differences in the resulting sea-ice freeboard between both sensors. However, these patterns of sea-ice freeboard differences are related to differences in the Envisat waveform parameters of sea-ice backscatter and leading-edge width (Figures 2b-c/3b-c; as well as pulse peakiness, which is strongly correlated with sea-ice backscatter, but is not shown here). These waveform parameter variations in turn reflect changes in the surface properties. Areas of potential MYI near the Canadian Archipelago and areas influenced by MYI export are in general substantially thinner for Envisat that CryoSat-2 (e.g., about 20 cm and more in March, Figure 3a). On the other side, areas of predominantly FYI are in general thicker in the Envisat data (Figure 2a). However, the level of freeboard difference is not constant throughout a winter season but rather appears to be seasonal, where Envisat appears to be unable to keep track of these seasonal changes.

As these differences in sea-ice freeboard between CryoSat-2 and Envisat appear to be indeed strongly correlated to patterns in the sea-ice backscatter and the leading-edge width of Envisat waveforms (Figures 2b-c/3b-c), we decided to apply a novel



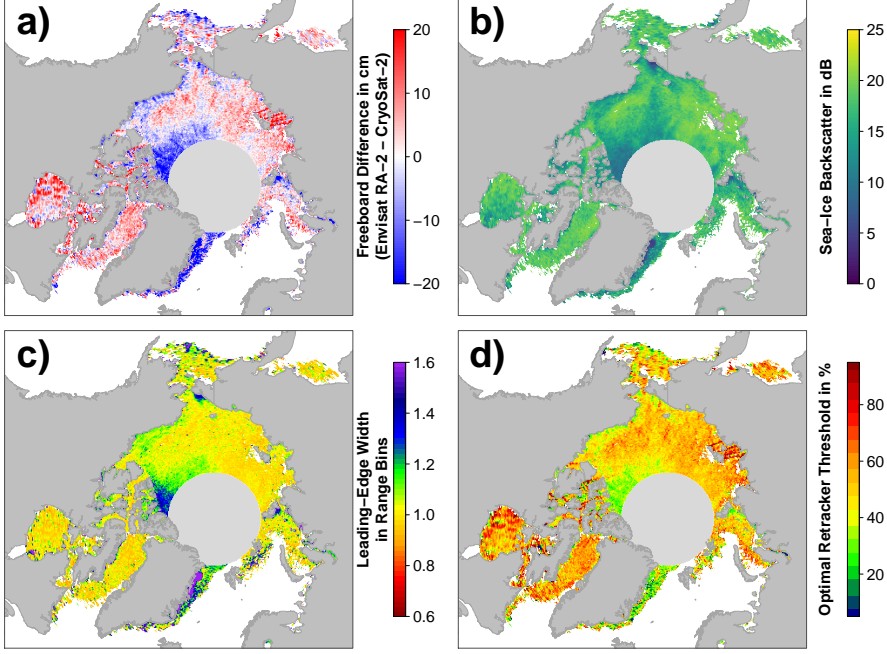

**Figure 3.** Exemplary visualizations for the Arctic in March 2012 in the same setup as Figure 2.

empirical tuning scheme by computing an adaptive range-retracker threshold as a function of sea-ice backscatter and the leading-edge width to mitigate the differences. Due to the already mentioned larger footprint of Envisat and hence increased mixing of different surface types, it appears to be necessary to treat waveforms differently according to the wave-form shape (and hence surface properties) by means of retracking the main scattering horizon. Guerreiro et al. (2017) proposed in their study a correction scheme deriving a relationship between monthly pulse peakiness and the monthly freeboard difference between CryoSat-2 and Envisat based on a third order polynomial fit. In contrast to applying a similar post-retracking correction to the resulting freeboard estimates, we apply our correction already during waveform retracking.

In order to derive a functional relationship between retracker threshold and sea-ice backscatter/leading-edge width, we first processed all available Envisat data for the complete MOP. This processing was done using the TFMRA with a fixed threshold for leads of 95 % and a threshold for sea-ice-type waveforms that was changed in each run. This sea-ice threshold ranged between 5 % and 95 % in steps of 5 %. For example, in the first run the complete data set was processed using a retracker threshold of 5 % for sea-ice-type waveforms and the resulting sea-ice freeboard was calculated. In the next run, a fixed threshold of 10 % was used for all sea-ice-type waveforms and so on. This continued until the last run was computed with a retracker threshold of 95 % for sea-ice-type waveforms and the resulting sea-ice freeboard was calculated.

From this data set, the optimal threshold, i.e., the threshold that yields the smallest possible difference in sea-ice freeboard between Envisat and CryoSat-2, was iteratively derived. Exemplary results are shown in Figures 2d and 3d. Again, the seasonal



change observed in the waveform parameters is also reflected in the resulting optimal threshold values. These show a varying range of optimal-threshold values that are in general higher for the early winter compared to the late winter.

Next, average optimal-threshold values were calculated for each bin in steps of $0.25\,\mathrm{dB}$ for sea-ice backscatter and $0.025$ for the leading-edge width on a x-y plane. The months November through March are covered twice during the mission-overlap

period (MOP) and both occurrences were used. October and April, which were only covered once during the MOP, were each added twice to circumvent issues of under-representation in their number of data values added to the total.

Through this compilation of monthly data points, three third-order polynomial planes were fitted based on different weighting schemes in order to maximize the adjusted coefficient of determination ($R^2_{adj}$). $R^2_{adj}$ is a measure for the quality of the model fit. In contrast to the normal $R^2$, $R^2_{adj}$ decreases through adding useless predictors to a model and is therefore a more

robust measure for model quality than $R^2$. As weights, we either used the number of optimal-threshold values per bin in the x-y plane, the inverse standard deviation of all optimal threshold values per bin ($1/\sigma$), or no weights at all.

The optimal threshold ($th_{opt}$; in decimal values) to be used in the adaptive range retracking for the Arctic as a function of sea-ice backscatter ($\sigma^o$) and leading-edge width ($lew$) is given by Equation 2:

$$
\begin{aligned}
th_{opt} = {}& 3.4775697362 \\
& - 5.9296875486 \times lew + 4.3516498381 \times lew^2 \\
& - 1.0933131955 \times lew^3 - 0.0914747272 \times \sigma^o \\
& + 0.0063983796 \times \sigma^{o2} - 0.0001237455 \times \sigma^{o3}
\end{aligned}
\tag{2}
$$

For the Arctic, Equation 2 achieved the highest adjusted $R^2_{adj}$ of 0.94 with the inverse standard deviation as weights. All used data points have a minimum of 50 occurrences to reduce noise and were obtained in the central Arctic only (i.e., we

excluded areas such as the Canadian Arctic Archipelago and the Hudson Bay, but also extensive fast-ice areas such as in the Laptev Sea).

In a first attempt, we applied Equation 2 also to the southern hemisphere. However, this did not result in an improvement of the freeboard differences between Envisat and CryoSat-2. The reason for that can partly be seen in Figures 4 and 5.

In contrast to the Arctic (Figures 2 and 3), there is less seasonality in the data, i.e., the differences between early and late

winter are less prominent in the sea-ice freeboard differences as well as the optimal-threshold values. Overall, a wider range of optimal-threshold values is necessary at any given month, in order to achieve a minimum freeboard difference (Figures 4d and 5d).

Additionally, the overall range and distribution of the sea-ice backscatter is different between the Arctic and the Antarctic (not shown) as well as patterns in sea-ice backscatter and leading-edge width are less correlated in some areas (Figures 4b-c

and 5b-c). This is potentially related ice-snow interface flooding paired with subsequent refreezing and formation of snow ice, large fast ice areas with a different snow stratigraphy and depth, and a different ice-growth history than in the Arctic causing a larger fraction of rough and deformed sea ice.



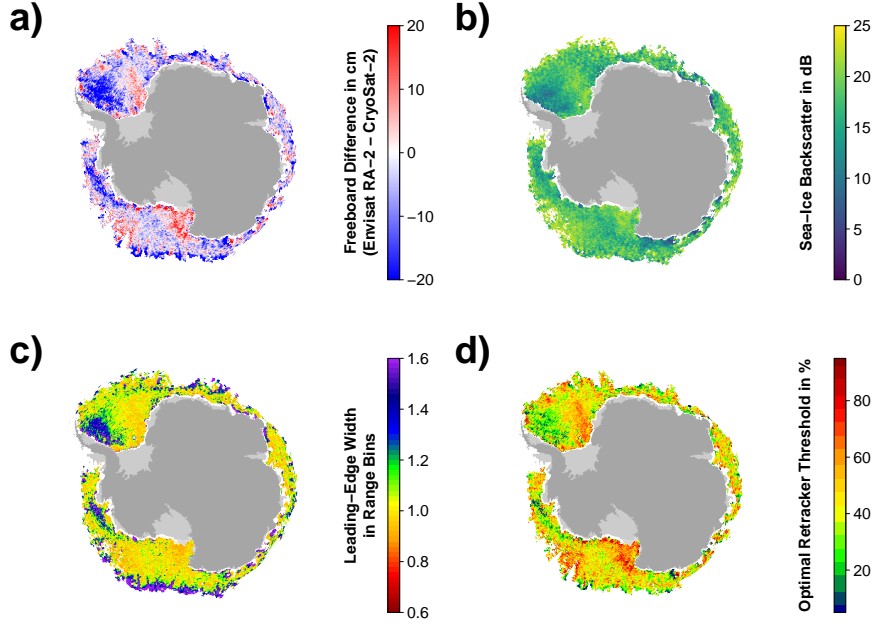

**Figure 4.** Exemplary visualizations for the Antarctic in May 2011 in the same setup as Figure 2.

ice-snow interface flooding paired with subsequent refreezing and formation of snow ice, large fast ice areas with a different snow stratigraphy and depth, and a different ice-growth history than in the Arctic causing a larger fraction of rough and deformed sea ice.

For the Antarctic, a second-order polynomial fit resulted in the best statistical result ($R^2_{adj}$ of 0.77) to describe the optimal threshold as a function of leading-edge width and sea-ice backscatter. Equation 3 summarizes the relationship for deriving the optimal threshold ($th_{opt}$; in decimal values) in the Antarctic as a function of sea-ice backscatter ($\sigma^o$) and leading-edge width ($lew$):

$$
\begin{aligned}
th_{opt} = {} & 0.8147895184 \\
& - 0.5555823623 \times lew + 0.1347526920 \times lew^2 \\
& + 0.0055934198 \times \sigma^o - 0.0001431595 \times \sigma^{o2}
\end{aligned}
\tag{3}
$$

Here, the best fit is obtained using the total number of optimal-threshold values per bin as weights. All used data points also have a minimum of 50 occurrences and were obtained by excluding ice zones around the Antarctic that appear to be influenced by ocean swell (identified by sea-ice backscatter and/or leading edge with outlier artifacts) as well as the months from December through April.



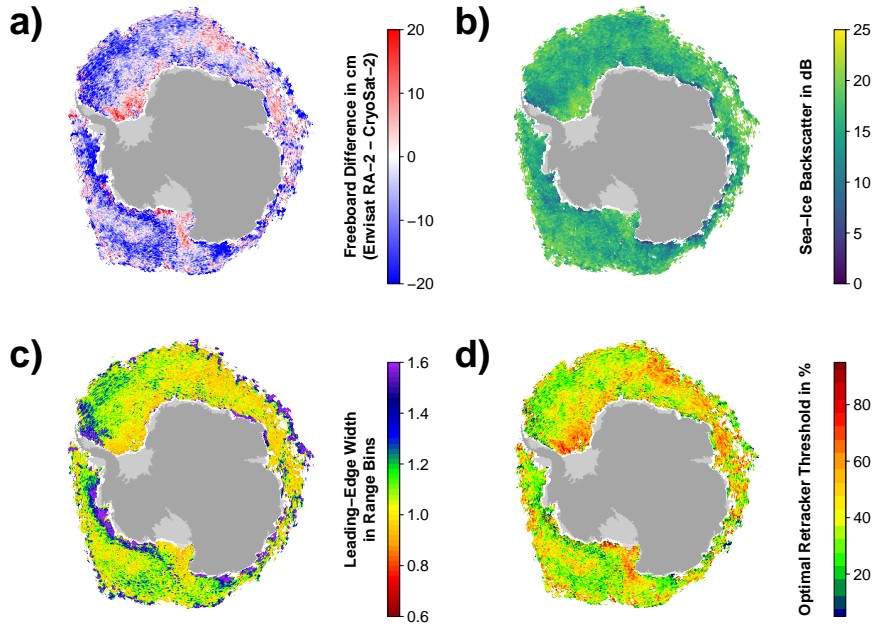

**Figure 5.** Exemplary visualizations for the Antarctic in September 2011 in the same setup as Figure 2.

Utilizing both equations, for each range-retracking of every sea-ice waveform, the to-be-used threshold is calculated from the waveform-associated sea-ice backscatter and leading-edge width value. This threshold is then believed to yield the mean-scattering surface in best accordance to the CryoSat-2 measurements.

## 3 Results and Discussion

5 In this section we want to present and discuss the results obtained for the mission overlap period (MOP) between Envisat and CryoSat-2. This is presented first for the surface-type classification and then for the range retracking and the associated sea-ice freeboard retrieval.

### 3.1 Surface-type classification

Utilizing our surface-type classification scheme results in an overall much better agreement between CryoSat-2 and Envisat 10 based on lead-, sea-ice, and valid fractions (Figures 6 – 9). Compared to the surface-type classification used during SICCI-1 for Envisat, our approach is less strict and allows for substantially more waveforms being classified as either lead or sea-ice type that were rejected before. Additionally, a very high fraction of lead detections was present, compared to only a very small fraction of classified sea-ice type waveforms during SICCI-1 (Schwegmann et al., 2016). Furthermore, the inter-mission





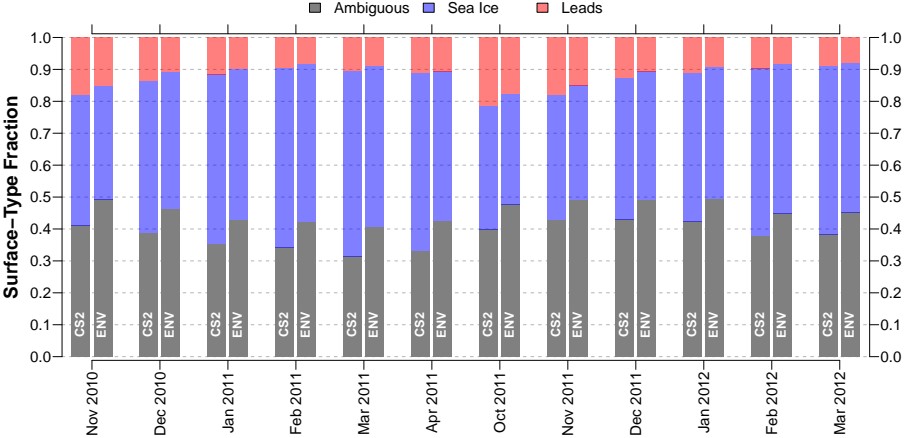

**Figure 6.** Time-series of surface-type fractions for the mission overlap period between CryoSat-2 (CS2) and Envisat (ENV) for the Arctic based on orbit-track (i.e., non gridded) data.

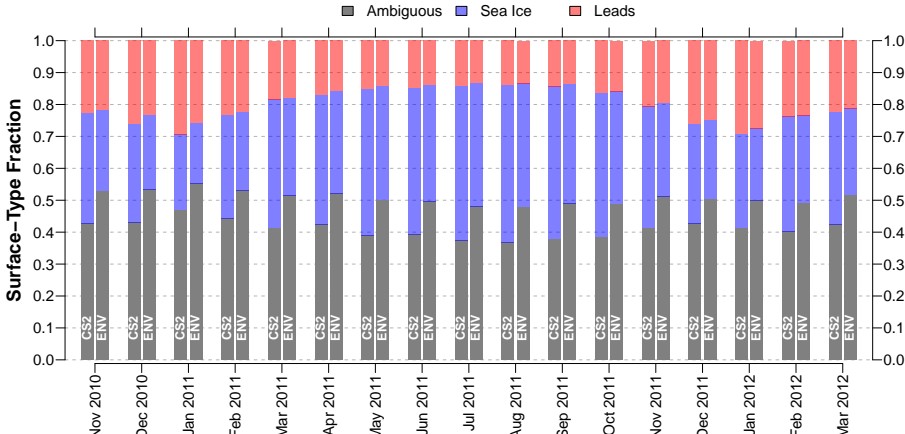

**Figure 7.** Time-series of surface-type fractions for the mission overlap period between CryoSat-2 (CS2) and Envisat (ENV) for the Antarctic based on orbit-track (i.e., non gridded) data.

consistency of the surface-type classification for the Arctic as well as the Antarctic has improved substantially (Figures 8 and 9).

The increased number of valid waveforms has an additional positive side effect on the overall data record: It allows for a much finer grid resolution to be used in the final Level 3 product without any concessions on overall coverage. Here, we are

5   now able to provide a $25\,\mathrm{km} \times 25\,\mathrm{km}$ ($50\,\mathrm{km} \times 50\,\mathrm{km}$) resolution gridded data set for the Arctic (Antarctic) compared to the $100\,\mathrm{km} \times 100\,\mathrm{km}$ during SICCI-1.





Direct comparisons of surface-type class fractions (i.e., either ambiguous, lead, or sea-ice type) over the course of the MOP reveal an overall good agreement between CryoSat-2 and Envisat based on the non-gridded orbit data (Figures 6 and 7). While the fractions of lead- and sea-ice waveforms are on average slightly smaller for Envisat than for CryoSat-2 (about 8 % for the Arctic and 10 % for the Antarctic), both sensors show a similar seasonal development in both hemispheres. Especially,

the fraction differences of detected leads are very small with a root-mean-squared difference (RMSD) of 2 % and 1 % for the Arctic and Antarctic, respectively. The discrepancy is larger for detected sea-ice waveforms with a RMSD of 6 % for the Arctic and 9 % for the Antarctic.

Exemplary visualizations of monthly gridded inter-comparisons between Envisat and CryoSat-2 based on valid-, lead-, and sea-ice waveform fractions are shown in Figures 8 and 9. In these gridded data sets, the overall good agreement is confirmed.

On average, the gridded valid waveform fraction for Envisat is 9 % (11 %) lower in the Arctic (Antarctic) than the ones achieved by CryoSat-2. This behavior is expected in regions with high rates of sea-ice dynamics such as the Beaufort Sea, where the increased surface-type mixing from the much larger footprint of Envisat likely prevents a clearer separation between waveform types. Lead- and sea-ice fractions differ by 2 % and 4 % for the Arctic and Antarctic, respectively.

Nevertheless, both comparisons highlight the overall good agreement that could be achieved between both sensors with this

inter-mission consistent surface-type classification scheme. These results therefore lay the foundation for a proper inter-mission sea-ice freeboard and sea-ice thickness data record.

## 3.2 Range retracking and freeboard retrieval

In this subsection, we show and discuss the results of using the adaptive threshold retracker for Envisat. For the Arctic, Figure 10 shows the histograms of CryoSat-2 (blue) and Envisat (red) freeboards in centimeters as well as the histogram of the

resulting freeboard differences (Envisat minus CryoSat-2; gray). Furthermore, average freeboard estimates for all distributions per month during the mission-overlap period for Envisat and CryoSat-2 are shown.

While in the first winter season, the match is nearly perfect with absolute average freeboard differences below one centimeter, differences during the second winter season increase up to about three centimeters. Nevertheless, this is a substantial improvement over any previous comparisons conducted during SICCI-1. Especially for the Arctic spring period (March and

April), differences in average freeboard are 1.2 cm or less. The maximum monthly average freeboard differences is 2.2 cm.

A comparison to the results of Guerreiro et al. (2017) is constrained by the different methods used. Besides the pulse-peakiness correction of the Envisat monthly freeboard, Guerreiro et al. (2017) also apply a 25 km along-track median smoothing to all freeboard estimates and afterward discard all freeboard estimates below -1 m and above 2 m, respectively. The resulting values are then used to compile the monthly gridded data set. In this study, we set the lower and upper sea-ice free-

board thresholds to -0.25 m and 2.25 m, respectively, without applying any smoothing. Values outside this range are discarded. Furthermore, we also compute freeboard results outside the central Arctic basin and take them into account for our comparison. While the differences between CryoSat-2 and Envisat freeboard appear to be comparable between both studies, the average monthly sea-ice freeboard estimates for both sensors are between about 3 cm and 10 cm larger in our study compared to the results shown in Guerreiro et al. (2017).



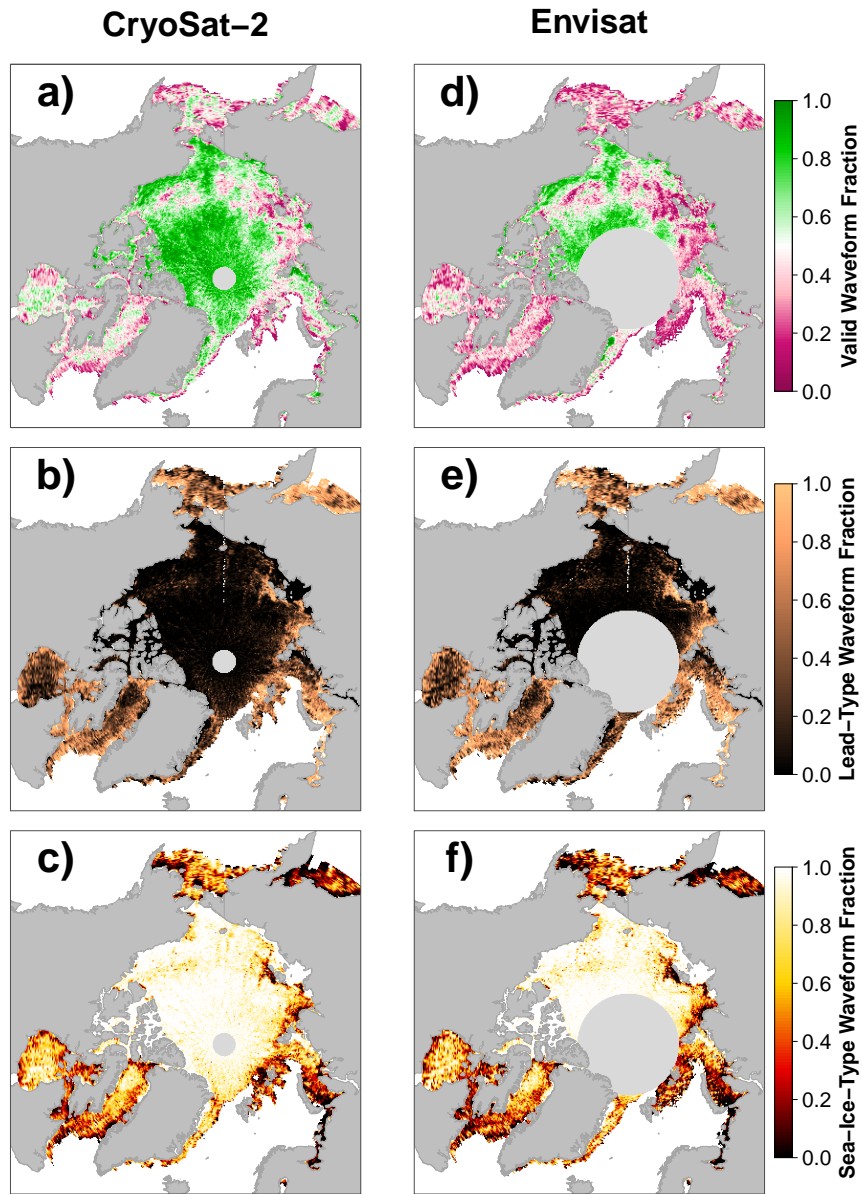

**Figure 8.** Comparison of gridded surface-type classification benchmarks of valid waveform fractions (a/d; ratio of either lead-type/sea-ice-type classification to the total number of waveforms per grid cell), lead-type waveform fraction (b/e; ratio of lead-type classifications to the number of valid classifications per grid cell), and the sea-ice-type waveform fraction (c/f; ratio of sea-ice-type classifications to the number of valid classifications per grid cell) between CryoSat-2 and Envisat for March 2012 in the Arctic.

For the Antarctic, results are not as good as for the Arctic (Figure 11). Overall, the approach has less skill to match Envisat sea-ice freeboards to the ones of CryoSat-2. This is very likely related to other physical process such as a more complex snow





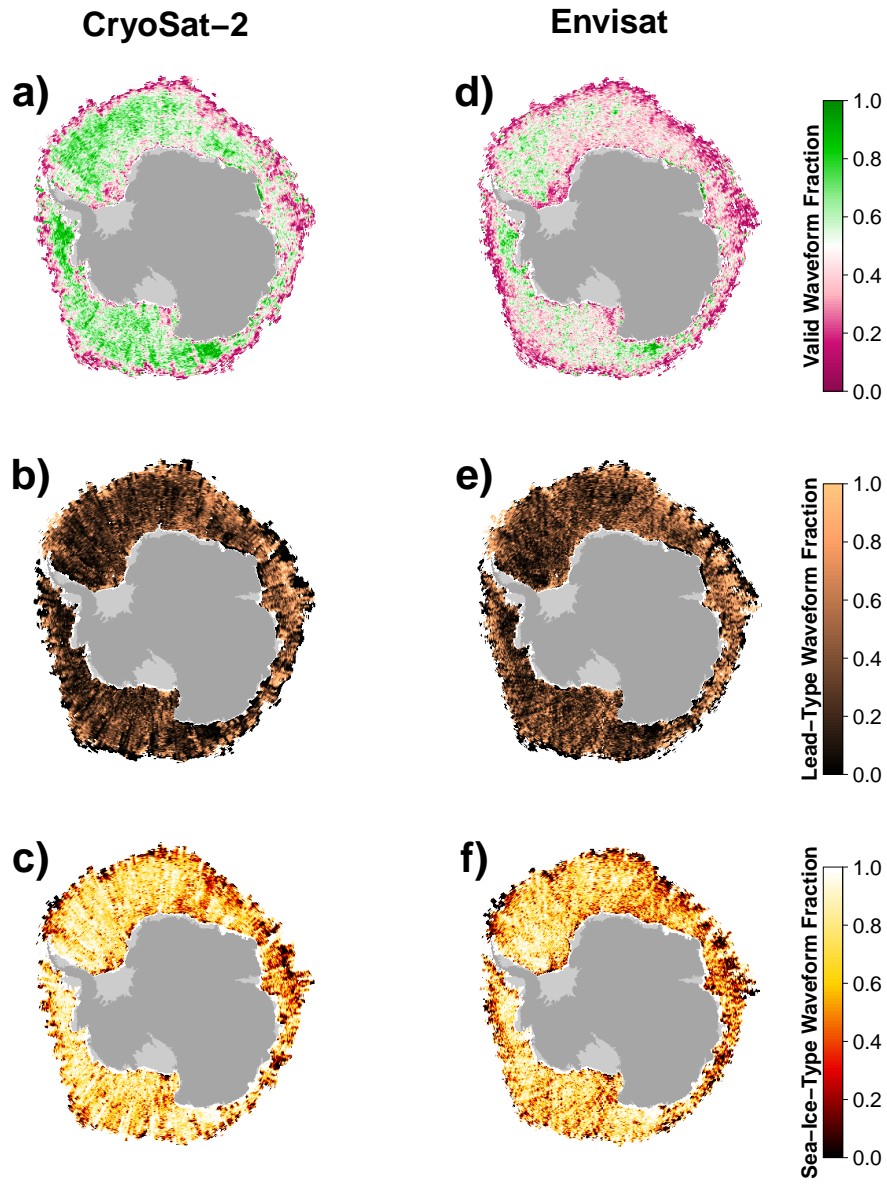

**Figure 9.** Comparison of gridded surface-type classification benchmarks in the same setup as Figure 8 between CryoSat-2 and Envisat for September 2011 in the Antarctic.

stratigraphy caused by the more strongly varying weather patterns with melt-refreeze cycles even in the middle of winter. Furthermore, snow/ice-interface flooding causes a (temporarily) wet and saline basal snow layer and influences the sea-ice surface roughness. However, issues causing these sensor differences are subject to further investigation. Overall, there is a





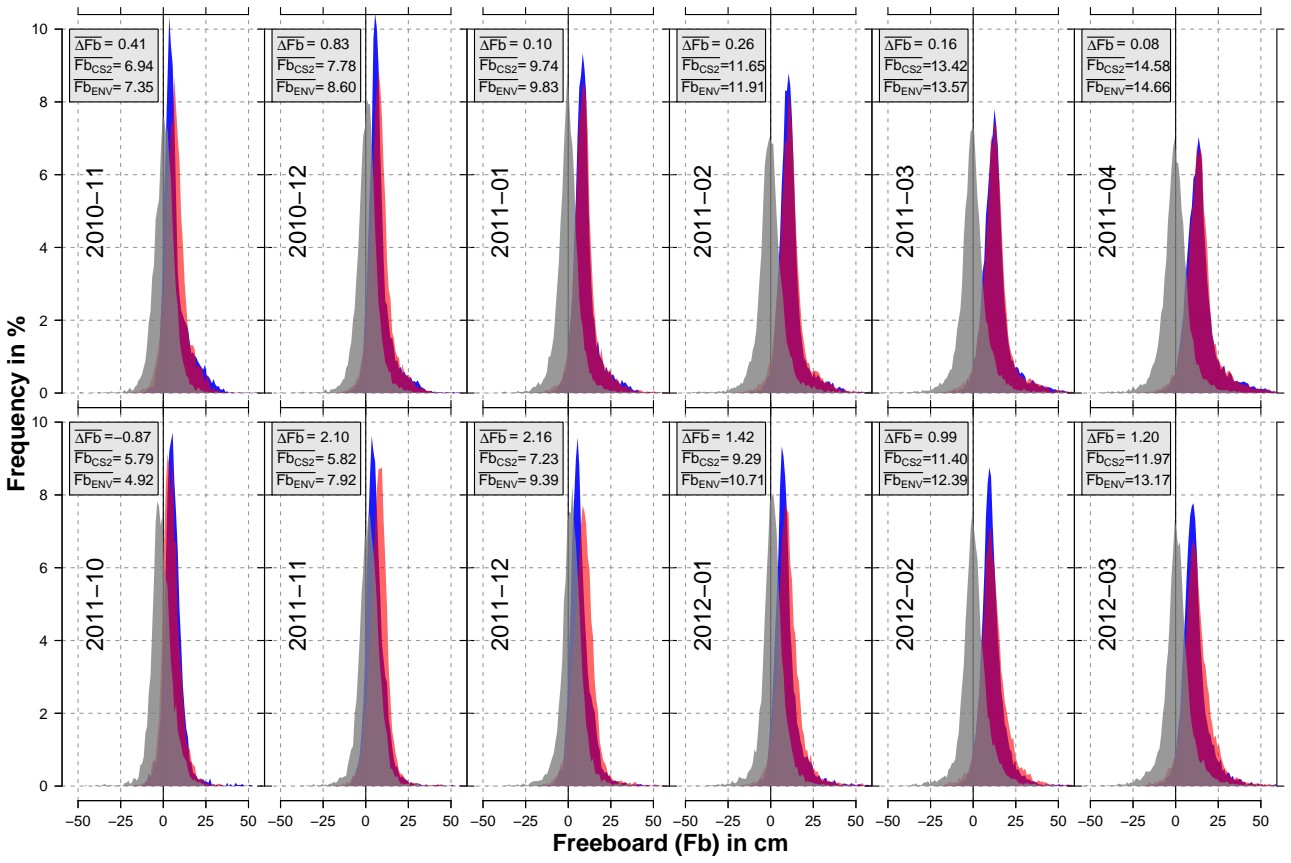

**Figure 10.** Histograms of freeboard for each month of the mission overlap period (in centimeters) for Envisat (red) and CryoSat-2 (blue) as well as the corresponding freeboard difference between both sensors (gray) for the Arctic. Furthermore, the average freeboard difference ($\Delta \bar{F}b$; Envisat minus CryoSat-2), as well as the average freeboards for CryoSat-2 ($\bar{Fb}_{CS2}$) and Envisat ($\bar{Fb}_{ENV}$) are given in centimeters in the gray box for each month.

stronger seasonality in the monthly freeboard differences between summer and winter, which also leads towards a higher maximum monthly average freeboard difference of about 2.7 cm.

A different way in visualizing these results is shown in Figure 12. Here, the cumulative frequencies of absolute freeboard differences (Envisat minus CryoSat-2, i.e. the gray histograms in Figures 10 and 11) are averaged over the complete MOP for

5 both hemispheres. For the Arctic (Figure 12a), more than 50 % of all data points are in an absolute freeboard difference range of ±4 cm. Furthermore, more than 75 %(90 %) of all data points are in a range of ±7 cm(±12 cm) absolute freeboard difference. However, for the Antarctic (12b) results are not as good. In order to achieve values of 50 %,75 %, and 90 % cumulative frequency, absolute freeboard difference increase to ±7 cm, ±12 cm,and ±20 cm, respectively.

Resulting freeboard differences using our new approach for Envisat and CryoSat-2 are shown in Figure 13 for the Arctic

10 and in Figure 14 for the Antarctic. Shown are the same months as in Figures 2 through 5. While the overall differences are





**Figure 11.** Histograms of freeboard for each month of the mission overlap period (in centimeters) for Envisat (red) and CryoSat-2 (blue) as well as the corresponding freeboard difference between both sensors (gray) for the Antarctic. Furthermore, the average freeboard difference ($\Delta \bar{Fb}$; Envisat minus CryoSat-2), as well as the average freeboards for CryoSat-2 ($\bar{Fb}_{CS2}$) and Envisat ($\bar{Fb}_{ENV}$) are given in centimeters in the gray box for each month.





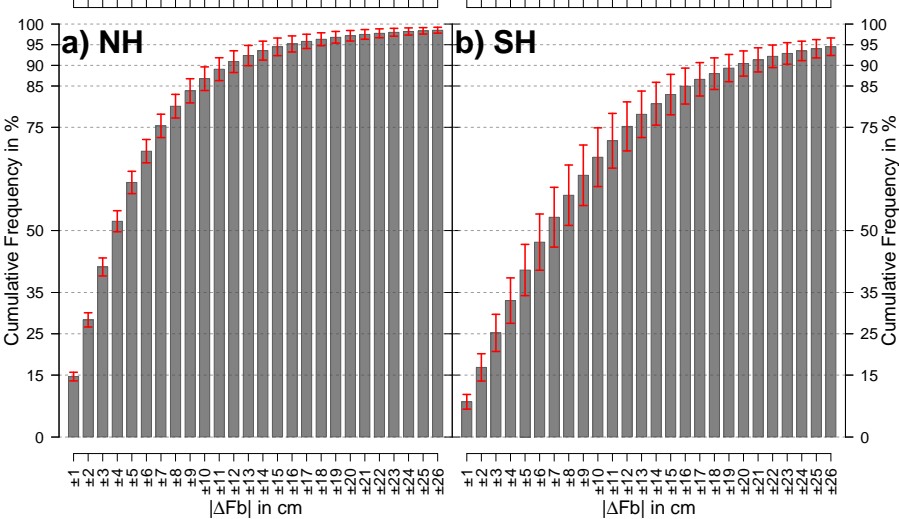

**Figure 12.** Cumulative frequencies of absolute freeboard differences (Envisat minus CryoSat-2) as shown in Figures 10 and 11 averaged over all months in the MOP. Data is presented for the northern hemisphere (a) as well as the southern hemisphere (b). Errorbars indicate $\pm 1$ standard deviation. First bar covers the absolute freeboard difference range from [0cm:1cm[, second bin from [1cm:2cm[, and so on.

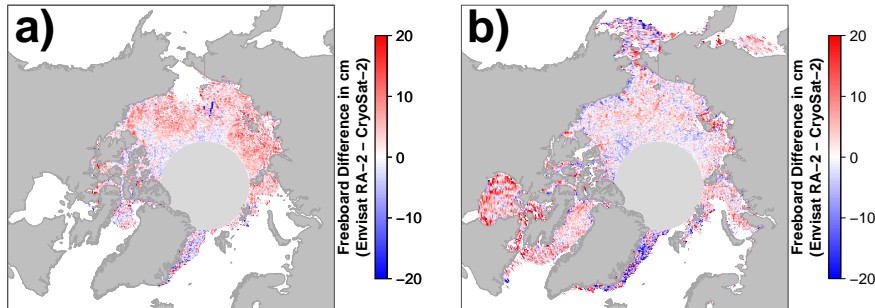

**Figure 13.** Visualizations of freeboard differences between Envisat using our new approach and CryoSat-2 in November 2011 (a) and March 2012 (b) for the Arctic (compare Figures 2a and 3a).

minimized, especially in the Arctic, there are still areas with rather large freeboard differences. In the Arctic, these comprise the Hudson Bay and the area east of Greenland. In the Antarctic, while the differences are lowered, the overall differences remain larger.



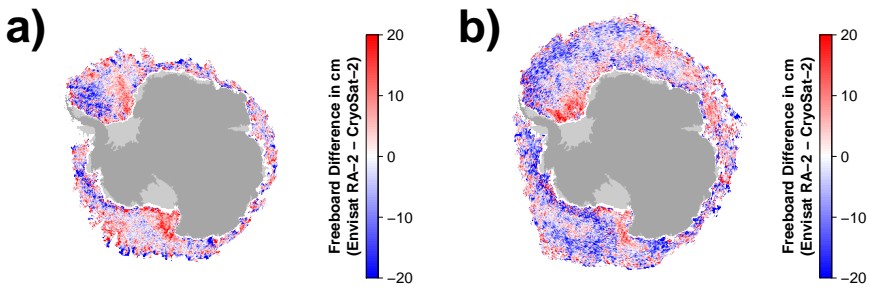

**Figure 14.** Visualizations of freeboard differences between Envisat using our new approach and CryoSat-2 in May 2011 (a) and September 2011 (b) for the Antarctic (compare Figures 4a and 5a).

## 4   Summary and Outlook

This study showed the potential of a combined novel surface-type classification scheme in combination with a waveform-parameter dependent adaptive threshold retracker approach in order to create a consistent data set of Envisat and CryoSat-2 sea-ice freeboard estimates. This approach is based on the observed correlation between freeboard differences between both sensors and the waveform characteristics of sea-ice backscatter and leading-edge width. Their spatio-temporal variations in acquired Envisat waveforms reflect changes in surface properties such as the surface roughness and footprint-size dependent surface-type mixing. We applied this approach for the mission overlap period from November 2010 to March 2012 and then used it to iteratively train and apply an adaptive threshold retracker to Envisat for both hemispheres. Different sea-ice conditions in both hemispheres also result in different inter-mission biases between Envisat and CryoSat-2. In contrast to previous attempts during SICCI-1, the inter-mission sea-ice freeboard biases could be minimized.

Furthermore, through the application of our inter-mission consistent surface-type classification in SICCI-2, a much higher comparability between the amount and location of positively identified lead-type and sea-ice-type waveforms for Envisat and CryoSat-2 could be achieved. Additionally, due to the higher amount of identified waveforms, a resolution of 25 km × 25 km and 50 km × 50 km could be realized in the final gridded data product for the Arctic and Antarctic, respectively. This is a substantial improvement over SICCI-1, where due to the much stricter surface-type classification and associated low identification/classification rates of waveforms, a resolution of only 100 km × 100 km could be achieved without a substantial drop in spatial data coverage for Envisat.

The next step is to start creating an inter-mission consistent and reliable climate data record on Arctic and Antarctic sea-ice thickness and volume as well as to investigate inter-annual as well as seasonal changes. Moreover, investigation of the stability of assumptions in auxiliary data sets is necessary. Here, especially the validity of the used snow-depth climatology in the freeboard to thickness conversion in a changing Arctic needs to be investigated.



*Author contributions.*  Stephan Paul designed the surface-type classification as well as the adaptive retracker approach, conducted the analysis and drafted the original manuscript. Stefan Hendricks developed pySIRAL, assisted with the analysis, the approach development as well as the data processing using pySIRAL. Robert Ricker, Stefan Kern, and Eero Rinne assisted in the analysis as well as in writing the manuscript.

*Competing interests.*  The authors declare no conflict of interests.

5  *Acknowledgements.*  This work was funded through the ESA Climate Change Initiative. The authors would like to thank the International Space Science Institute (ISSI Bern) for its support to an international team focused on the current scientific issues in the observation of the sea level and sea ice at polar latitudes, which fostered fruitful discussion on the ideas for this paper.



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

## Appendix A:  Surface-Type Classification Thresholds





**Table A1.** Pulse-peakiness thresholds for the surface-type classification of lead-type waveforms for Envisat, CryoSat-2 SAR mode, and CryoSat-2 SIN mode data for the Arctic and the Antarctic. Values for the pulse peakiness are unit-less.

| Month | Envisat | | | | CryoSat-2 SAR | | | | CryoSat-2 SIN | | | |
| | Arctic | | Antarctic | | Arctic | | Antarctic | | Arctic | | Antarctic | |
| | Min | Max | Min | Max | Min | Max | Min | Max | Min | Max | Min | Max |
|---|---|---|---|---|---|---|---|---|---|---|---|---|
| Jan | 46.90 | – | 56.60 | – | 67.30 | – | 80.70 | – | 264.30 | – | 307.40 | – |
| Feb | 46.40 | – | 53.20 | – | 66.30 | – | 75.10 | – | 257.90 | – | 300.70 | – |
| Mar | 46.20 | – | 51.90 | – | 66.60 | – | 73.20 | – | 253.60 | – | 291.70 | – |
| Apr | 48.40 | – | 50.70 | – | 69.90 | – | 69.50 | – | 264.60 | – | 288.50 | – |
| May | – | – | 50.10 | – | – | – | 69.70 | – | – | – | 283.70 | – |
| Jun | – | – | 49.30 | – | – | – | 69.30 | – | – | – | 284.20 | – |
| Jul | – | – | 49.50 | – | – | – | 69.20 | – | – | – | 276.90 | – |
| Aug | – | – | 49.10 | – | – | – | 69.50 | – | – | – | 284.40 | – |
| Sep | – | – | 49.30 | – | – | – | 69.70 | – | – | – | 278.90 | – |
| Oct | 52.90 | – | 51.60 | – | 76.00 | – | 71.70 | – | 291.80 | – | 289.40 | – |
| Nov | 51.00 | – | 53.90 | – | 73.80 | – | 76.00 | – | 288.80 | – | 299.40 | – |
| Dec | 47.70 | – | 55.10 | – | 68.60 | – | 78.10 | – | 272.60 | – | 307.70 | – |

**Table A2.** Sea-ice backscatter thresholds for the surface-type classification of lead-type waveforms for Envisat, CryoSat-2 SAR mode, and CryoSat-2 SIN mode data for the Arctic and the Antarctic. Values for sea-ice backscatter are given in dB.

| Month | Envisat | | | | CryoSat-2 SAR | | | | CryoSat-2 SIN | | | |
| | Arctic | | Antarctic | | Arctic | | Antarctic | | Arctic | | Antarctic | |
| | Min | Max | Min | Max | Min | Max | Min | Max | Min | Max | Min | Max |
|---|---|---|---|---|---|---|---|---|---|---|---|---|
| Jan | 28.80 | – | 33.20 | – | 23.80 | – | 28.50 | – | 24.90 | – | 29.20 | – |
| Feb | 28.60 | – | 32.10 | – | 23.20 | – | 26.80 | – | 25.00 | – | 29.00 | – |
| Mar | 28.50 | – | 31.80 | – | 23.30 | – | 26.20 | – | 24.10 | – | 28.50 | – |
| Apr | 28.40 | – | 30.80 | – | 23.40 | – | 24.60 | – | 24.50 | – | 27.80 | – |
| May | – | – | 29.40 | – | – | – | 23.40 | – | – | – | 26.90 | – |
| Jun | – | – | 28.60 | – | – | – | 22.80 | – | – | – | 26.50 | – |
| Jul | – | – | 28.60 | – | – | – | 23.00 | – | – | – | 26.30 | – |
| Aug | – | – | 28.40 | – | – | – | 23.00 | – | – | – | 27.00 | – |
| Sep | – | – | 28.50 | – | – | – | 23.20 | – | – | – | 26.20 | – |
| Oct | 32.80 | – | 29.50 | – | 28.00 | – | 24.00 | – | 29.00 | – | 27.20 | – |
| Nov | 30.80 | – | 31.10 | – | 25.80 | – | 25.90 | – | 27.40 | – | 27.50 | – |
| Dec | 29.30 | – | 32.10 | – | 24.10 | – | 27.30 | – | 25.80 | – | 28.40 | – |



**Table A3.** Leading-edge width thresholds for the surface-type classification of lead-type waveforms for Envisat, CryoSat-2 SAR mode, and CryoSat-2 SIN mode data for the Arctic and the Antarctic. Values are in range-bin fractions for the leading-edge width.

| Month | Envisat | | | | CryoSat-2 SAR | | | | CryoSat-2 SIN | | | |
| | Arctic | | Antarctic | | Arctic | | Antarctic | | Arctic | | Antarctic | |
| | Min | Max | Min | Max | Min | Max | Min | Max | Min | Max | Min | Max |
|---|---|---|---|---|---|---|---|---|---|---|---|---|
| Jan | – | 0.82 | – | 0.82 | – | 0.77 | – | 0.71 | – | 1.10 | – | 1.00 |
| Feb | – | 0.82 | – | 0.82 | – | 0.78 | – | 0.73 | – | 1.11 | – | 1.01 |
| Mar | – | 0.82 | – | 0.82 | – | 0.78 | – | 0.74 | – | 1.13 | – | 1.03 |
| Apr | – | 0.82 | – | 0.82 | – | 0.76 | – | 0.77 | – | 1.09 | – | 1.04 |
| May | – | – | – | 0.82 | – | – | – | 0.77 | – | – | – | 1.06 |
| Jun | – | – | – | 0.82 | – | – | – | 0.77 | – | – | – | 1.05 |
| Jul | – | – | – | 0.82 | – | – | – | 0.78 | – | – | – | 1.07 |
| Aug | – | – | – | 0.82 | – | – | – | 0.77 | – | – | – | 1.05 |
| Sep | – | – | – | 0.82 | – | – | – | 0.77 | – | – | – | 1.07 |
| Oct | – | 0.82 | – | 0.82 | – | 0.72 | – | 0.76 | – | 1.02 | – | 1.05 |
| Nov | – | 0.82 | – | 0.82 | – | 0.73 | – | 0.74 | – | 1.03 | – | 1.02 |
| Dec | – | 0.82 | – | 0.82 | – | 0.76 | – | 0.72 | – | 1.07 | – | 1.00 |

**Table A4.** Pulse-peakiness thresholds for the surface-type classification of sea-ice-type waveforms for Envisat, CryoSat-2 SAR mode, and CryoSat-2 SIN mode data for the Arctic and the Antarctic. Values for the pulse peakiness are unit-less.

| Month | Envisat | | | | CryoSat-2 SAR | | | | CryoSat-2 SIN | | | |
| | Arctic | | Antarctic | | Arctic | | Antarctic | | Arctic | | Antarctic | |
| | Min | Max | Min | Max | Min | Max | Min | Max | Min | Max | Min | Max |
|---|---|---|---|---|---|---|---|---|---|---|---|---|
| Jan | – | 16.00 | – | 24.60 | – | 30.50 | – | 40.10 | – | 99.40 | – | 138.40 |
| Feb | – | 14.80 | – | 20.70 | – | 28.70 | – | 35.30 | – | 94.20 | – | 126.10 |
| Mar | – | 14.10 | – | 19.60 | – | 28.10 | – | 32.90 | – | 89.90 | – | 124.90 |
| Apr | – | 14.20 | – | 18.80 | – | 28.50 | – | 30.20 | – | 90.00 | – | 127.30 |
| May | – | – | – | 17.50 | – | – | – | 28.70 | – | – | – | 122.20 |
| Jun | – | – | – | 16.90 | – | – | – | 28.90 | – | – | – | 121.00 |
| Jul | – | – | – | 16.60 | – | – | – | 28.10 | – | – | – | 114.90 |
| Aug | – | – | – | 16.10 | – | – | – | 28.00 | – | – | – | 115.80 |
| Sep | – | – | – | 16.30 | – | – | – | 28.40 | – | – | – | 114.30 |
| Oct | – | 19.40 | – | 18.10 | – | 35.40 | – | 29.60 | – | 114.40 | – | 121.20 |
| Nov | – | 19.30 | – | 20.70 | – | 34.90 | – | 34.10 | – | 113.90 | – | 126.50 |
| Dec | – | 16.90 | – | 22.80 | – | 31.90 | – | 36.60 | – | 103.80 | – | 135.20 |





**Table A5.** Sea-ice backscatter thresholds for the surface-type classification of sea-ice-type waveforms for Envisat, CryoSat-2 SAR mode, and CryoSat-2 SIN mode data for the Arctic and the Antarctic. Values for sea-ice backscatter are given in dB.

| Month | Envisat | | | | CryoSat-2 SAR | | | | CryoSat-2 SIN | | | |
| | Arctic | | Antarctic | | Arctic | | Antarctic | | Arctic | | Antarctic | |
| | Min | Max | Min | Max | Min | Max | Min | Max | Min | Max | Min | Max |
|---|---|---|---|---|---|---|---|---|---|---|---|---|
| Jan | – | 22.50 | – | 27.20 | – | 20.80 | – | 26.30 | – | 21.40 | – | 26.40 |
| Feb | – | 21.80 | – | 25.40 | – | 19.90 | – | 24.10 | – | 20.90 | – | 25.10 |
| Mar | – | 21.30 | – | 26.70 | – | 19.60 | – | 25.10 | – | 20.10 | – | 27.60 |
| Apr | – | 20.40 | – | 27.20 | – | 19.00 | – | 26.20 | – | 19.10 | – | 27.30 |
| May | – | – | – | 24.60 | – | – | – | 23.10 | – | – | – | 24.90 |
| Jun | – | – | – | 23.10 | – | – | – | 20.90 | – | – | – | 24.20 |
| Jul | – | – | – | 22.50 | – | – | – | 20.20 | – | – | – | 24.10 |
| Aug | – | – | – | 21.70 | – | – | – | 19.10 | – | – | – | 24.90 |
| Sep | – | – | – | 22.30 | – | – | – | 20.00 | – | – | – | 23.70 |
| Oct | – | 25.90 | – | 23.30 | – | 25.70 | – | 20.60 | – | 24.30 | – | 25.00 |
| Nov | – | 24.60 | – | 25.20 | – | 23.20 | – | 22.90 | – | 23.70 | – | 25.20 |
| Dec | – | 22.80 | – | 26.10 | – | 21.10 | – | 23.90 | – | 22.00 | – | 25.00 |

**Table A6.** Leading-edge width thresholds for the surface-type classification of sea-ice-type waveforms for Envisat, CryoSat-2 SAR mode, and CryoSat-2 SIN mode data for the Arctic and the Antarctic. Values are in range-bin fractions for the leading-edge width.

| Month | Envisat | | | | CryoSat-2 SAR | | | | CryoSat-2 SIN | | | |
| | Arctic | | Antarctic | | Arctic | | Antarctic | | Arctic | | Antarctic | |
| | Min | Max | Min | Max | Min | Max | Min | Max | Min | Max | Min | Max |
|---|---|---|---|---|---|---|---|---|---|---|---|---|
| Jan | 0.81 | – | 0.78 | – | 1.02 | – | 0.87 | – | 1.55 | – | 1.31 | – |
| Feb | 0.83 | – | 0.80 | – | 1.08 | – | 0.95 | – | 1.58 | – | 1.40 | – |
| Mar | 0.83 | – | 0.80 | – | 1.10 | – | 0.98 | – | 1.62 | – | 1.37 | – |
| Apr | 0.83 | – | 0.80 | – | 1.11 | – | 1.02 | – | 1.64 | – | 1.34 | – |
| May | – | – | 0.81 | – | – | – | 1.07 | – | – | – | 1.37 | – |
| Jun | – | – | 0.80 | – | – | – | 1.07 | – | – | – | 1.38 | – |
| Jul | – | – | 0.80 | – | – | – | 1.12 | – | – | – | 1.41 | – |
| Aug | – | – | 0.81 | – | – | – | 1.13 | – | – | – | 1.41 | – |
| Sep | – | – | 0.81 | – | – | – | 1.11 | – | – | – | 1.42 | – |
| Oct | 0.78 | – | 0.80 | – | 0.91 | – | 1.08 | – | 1.44 | – | 1.38 | – |
| Nov | 0.78 | – | 0.79 | – | 0.90 | – | 0.95 | – | 1.44 | – | 1.36 | – |
| Dec | 0.80 | – | 0.78 | – | 0.97 | – | 0.92 | – | 1.51 | – | 1.33 | – |