# Peer review of "Empirical Parametrization of Envisat Freeboard Retrieval of Arctic and Antarctic Sea Ice Based on CryoSat-2: Progress in the ESA Climate Change Initiative"

_The Cryosphere, 2018_

## Referee Comment (RC1) · T. W. K. Armitage (Referee) · 22 Mar 2018

Overview and general comments

The authors present algorithm development from the latest iteration of the ESA Sea Ice Climate Change Initiative, which is working toward producing a consistent long-term climate record of sea ice thickness for both polar oceans. The paper focusses on improving the surface classification methodology for both CryoSat-2 (CS2) and Envisat using multiple waveform parameters and more advanced statistical techniques, and then uses these results to tune the Envisat retracking threshold in order to minimize the observed freeboard difference relative to CS2. This is a somewhat similar approach to 'correcting' the Envisat sea ice freeboard as taken by Guerreiro et al.

[Figure]

(2017), where they adjusted the Envisat freeboard based on a fit between pulse peakiness and the CS2/Envisat freeboard difference. While there are studies similar to the present manuscript in the literature, the methodological development within the ESA SICCI warrants publication, as presumably further development will follow and eventually data will be delivered to users who will want to know how it was produced.

I do have some suggestions/comments that I think will improve the manuscript:

1) Sections 2.3.2 and 2.3.3: I am not familiar with these statistical techniques, and I don't think the wider polar altimetry community will be either, unless there have been other publications on these techniques within the context of polar altimetry? There is a lot of jargon (specialist terms e.g., "k-means", "clustering", "decision trees", "trained forests", etc.) and sentences that are nonsense without specialist knowledge, for example, how does a "decision tree" in a "trained forest" cast a "vote"?! I am imagining woodland creatures and some kind of election! While I appreciate that I could follow the references you have provided to textbooks etc, I think the manuscript would be greatly improved if you could provide a more intuitive explanation of these procedures, as well as some relevant equations and figures if applicable, including specific examples of how this applies to altimeter waveform classification. In particular, I think this is important because this seems to be one of the major developments in the paper, so I think readers should be able to gain an intuitive picture of what is happening in your processing in order that they might develop/reproduce what you have done. The schematic in Figure 1 is sort of useful but there is still a lot of jargon, and it's not clear to me what is happening at each stage.

2) How are your results affected by just using a fixed retracking threshold for Envisat lead waveforms, rather than including lead and floe waveforms in the tuning/fitting procedure? The reasoning for retracking near (or at) the maximum power for lead waveforms is equally valid for CS-2 and Envisat, i.e., specular scattering from leads reduces the effective footprint to the size of the lead, which in turn gives you a return which is close to the transmitted pulse (convolution with a delta function rather than the flatsurface impulse response). Using a single retracker for all CS2 waveforms and essentially two separate retrackers for Envisat leads and floes represents an inconsistency with your approach that I don't feel is justified.

3) Related to this, I appreciate that your fitting procedure is essentially try to match the Envisat freeboard to the CS2 freeboard by tuning the Envisat retracking threshold. But couldn't you simply skip a stage here and fit the Envisat waveform parameters (LEW/sigma-0) to the CS2 freeboard directly?

4) Section 3.1: You get a better match with the surface type classification than in SICCI-1, but isn't this by construction? Haven't you tried to match the number of waveforms classified as leads and floes for the two satellites, or have I misunderstood something? Further, I wouldn't necessarily expect there to be agreement between the number of leads/floes detected by the two instruments, simply because of the different footprints – from physical/geometric arguments I would expect far more 'ambiguous' waveforms in the Envisat data. More encouraging is the broad spatial agreement between the lead/floe distributions.

5) Section 3.2: Similarly, isn't the small observed difference in freeboard by construction? Here, I think the manuscript would be greatly improved by comparison of the two satellites with independent radar freeboard measurements, e.g., by combining the IceBridge laser and snow radar.

Specific comments

The title is clunky, I would suggest "Consistent retrievals of Arctic and Antarctic sea ice freeboard from Envisat and CryoSat-2"

Page 1, Line 2: I suggest "...estimation over recent years, however, precursor..."

Page 1, Line 13-15: "cover" should be "extent". Also, join the sentences "...Meier et al), while Antarctic sea ice extent..."

Page 1, Line 15-16: I suggest the following "...(Turner et al). Arctic sea ice is also

thinking, as observed by..."

Page 1, Line 24: I suggest "...that measurement of sea ice thickness at circumpolar scales in both polar regions..."

Page 2, Line 4: really this type of processing dates back to at least Laxon (1994), "Sea ice processing scheme at the EODC".

Page 2, Line 5: I'm not really familiar with the term "run-time" within the context of altimetry, could you explain or use a more familiar term.

Page 2, Line 6-7: "so accurate that...", this isn't really the case for individual lead/sea ice measurements due to speckle noise. Also, explain explicitly that this elevation difference is termed the freeboard, otherwise the next sentence might not make sense to people unfamiliar with this term.

Page 2, Line 9-11: "not true" – I'm aware of all the studies on this (including some of my own work!), but I would still argue that such a strong statement on this issue is still not possible. I would suggest "When estimating sea ice thickness from radar altimeters, it is often assumed that...", and you should provide a more extensive list of studies that might suggest otherwise.

Page 2, Line 18: I believe the Envisat altimeter was Radar Altimeter 2 (RA-2)?

Page 4, Line 32: Is this filtering important? How many waveforms are removed?

Page 4, Line 33-Page 5, Line 1: Do you also apply a land mask filter? Are the inbuilt land surface type flags accurate enough to catch all land contaminated waveforms?

Page 5, Line 4-8: I think what you are saying is that distinguishing leads is essential in order to estimate the instantaneous sea level anomaly along track?

Page 5, Line 9-14: Before this paragraph you should explain why it is possible to distinguish leads and floes to begin with i.e., explain the different surface scattering characteristics. Otherwise, this paragraph is not clear.

Page 5, Line 12: "footprint of 2km", this is the pulse-limited footprint. "increase up to 10km (Chelton et al.)", I don't think Chelton was talking about off-nadir ranging to leads, he was talking about the effect of significant wave height on the pulse limited footprint, which is fundamentally different i.e., strong off-nadir backscatter in the case of leads vs. large surface roughness in the case of high SWH.

Page 5, line 18: "sea ice backscatter", you mean $\sigma^0$?

Page 5, Line 29: young, thin ice areas, cause specular reflections, you should add a citation.

Page 5, Line 10-31: surely the rejection rate could be decreased as well?

Page 9, Line 24-Page 10, Line 3: This is all rather unclear to me.

Page 10, Line 1; Figures 2-5: Show the pulse peakiness maps as well, also show the multi-year ice mask for comparison.

Page 11, Line 2: Why disregard PP in the fitting procedure? Presumably you tried different iterations but discarding PP gave you the best result?

Page 12, Line 3-11: The fitting procedure is not clear to me at all. What are you fitting to what? What is the "x-y" plane? Perhaps a diagram/Figure could help to illustrate this?

Figure 10: Could you change the x-axis scale to ~ -25cm to ~ 40cm to make the figure clearer

Page 16, Line 26-34: Could the difference with Guerreiro be explained by the speed of light propagation correction, or do you both apply it in the same way?

---

## Referee Comment (RC2) · N. Kurtz (Referee) · 24 Mar 2018

This paper describes new procedures to provide consistency between sea ice freeboard records from Envisat and CryoSat-2. This is especially important due to the smaller effective footprint size of CryoSat-2 which needs to be accounted for in the retracking procedure as well as the classification of sea ice lead and floe echoes. The authors have done a good job to attempt to get as consistent record as possible which spans the two records, and much of what is done here is of significant value. But the work essentially treats CryoSat-2 data as a standard data set and adjusts the Envisat record to best match the record. As such, the CryoSat-2 data should be as accurate as possible to tie the records together. However, I did not see that this was adequately done in the paper. In particular I think there are flaws in the CryoSat-2 retracking

procedure which need to be addressed prior to publication, these are noted below. Particularly I note the need to better refine the lead tracking procedure and verify this through direct elevation comparisons between the measurements. Independent validation data of the CryoSat-2 data set through comparison with field campaigns such as CryoVEx or IceBridge are also needed.

Additionally, only freeboard differences are plotted so it is difficult to evaluate whether the retrieved freeboards themselves are accurate. Some maps and statistics of the actual retrieved freeboard are needed. This is especially important for the Antarctic region where due to the complexity of the surface prior studies with satellite radar altimeters have not demonstrated the capacity for obtaining accurate measurements.

Specific comments on the manuscript are given below:

P2, second paragraph: The wording choice is a bit awkward in parts...I'm not sure what quasi-nadir run-time measurement means here. "...which are so accurate" could be rephrased better.

P2 L18: "a the"

P4: If you have daily passive microwave measurements for snow depth retrievals then why is a climatology used for the Antarctic?

P5 L17-19: The use of SAR processing on CryoSat-2 will impact both the leading edge width as well as peakiness, I wouldn't expect these value to be equivalent to a pulse limited radar system for lead discrimination.

Section 2.3.2: Some further details on the k-means clustering is needed. Were the peakiness, leading edge width, and backscatter used here? What exactly is coming from the three clusters?

Section 2.3.3: Same here for the need for further details. What is the training data set that was used, and how was this selected? How was it clear that the method separated leads and floes other than the fact that they had expected values for peakiness

and backscatter? Was any validation done of the results to assess the quality of the classification?

Section 2.4: The need for different thresholds for sea ice leads and floes from CryoSat-2 was shown in Kurtz et al., 2014. This should apply to Envisat since both operate on the same physical principle: the effective geometrical area of the lead return is very small causing a radar return which is close to the transmit pulse shape. As both satellites have the same bandwidth the transmit pulse shape should be very similar for both CryoSat-2 and Envisat. However, for sea ice floes the pulse-limited footprint size of Envisat should require a different threshold than the unfocused SAR footprint of CryoSat-2. This implies the threshold chosen for CryoSat-2 floe returns needs to be adjusted. No matter the methodology used though, some validation of the choice of thresholds needs to be done and I think that is lacking in the manuscript.

Note too that the approach described in this section assumes the threshold used for CryoSat-2 is a control data set to which the Envisat data is tied, this means the threshold selected for the CryoSat-2 data set is of utmost importance. Thus some validation of this to demonstrate it is correct is sorely needed.

P11 L8-14: This test should be done on the retrieved elevations (not just freeboard values) between Envisat and CryoSat-2, particularly for leads. That would more clearly demonstrate whether the differences in the threshold algorithms are properly handled.

P11 L15: How was the optimal value chosen? Was it that which had the smallest mean difference, RMS difference, or something else?

Figures 10 and 13 seem to not match up visually. In Figure 13 there seems to be a far higher spatial coverage of red, indicating a higher Envisat freeboard whereas the distributions in Figure 10 seem to show only small mean differences and a more symmetric distribution. Some clarification on this is needed.

---

## Referee Comment (RC3) · S. FLEURY (Referee) · 28 Mar 2018

General Comments

In order to produce consistent freeboard from CryoSat-2 and Envisat, the authors outline the importance of having consistent leads and floes classification in one hand, and of having consistent waveform retracking in the other hand. The main difficulty of the task is due to the fact that the altimeters do not rely on the same technology, the traditional LRM altimeter for Envisat and the more recent SAR altimeter for CryoSat-2, and their radar echoes strongly differ.

For that purpose, they propose novel technics for these two main steps necessary to retrieve freeboard.

[Figure]

Regarding the classification, in order to overcome the rigidity of a fixe threshold on a unique parameter all over the year, they propose to rely on a combination of parameters, that can be computed on the waveforms for both altimeters: the pulse peakiness (pp), the backscatter (sig0) and the leading edge width (lew). The best criteria for the lead/floe classification is elaborated using a sequence of k-mean clustering followed by a random forest classification, and this for each of the months. In this approach, the first k-means clustering step is supposed to provide a ground truth for the next classification. Using this method, they produce statistically consistent lead/floe classifications between Envisat and CryoSat-2, and increase the number of "valid" waveforms in comparison with the SICCI-1 solution, which in turn allows increasing the resolution of the freeboard maps. However, if the consistency of the classification between the two satellite missions is an important prerequisite, it does not fully demonstrate the pertinence of the classification regarding the ground reality.

From the observation that a fixe threshold retracker is not able to retrieve a consistent freeboard distribution with a LRM altimeter, the authors propose to adapt this threshold using CryoSat-2 as a reference. The objective being to find out a law for the variable threshold according to two parameters of the waveforms that characterized the measured surface: the backscatter and the leading edge width. Thus, the philosophy is similar to the one suggested in Guerreiro et al (2017), but the method differs: this previous study uses fixe thresholds retrackers and computes a corrective law on the freeboard, from the discrepancies between Envisat and CryoSat-2 freeboards, according to the pulse peakiness to account for the surface roughness. In this new study, the threshold over the leads, as perceived by Envisat, is fixed whereas the threshold over the floes varies by step of 5% in order to find out the value that minimizes the freeboard differences with CryoSat-2, and this for each cell of each monthly grid. The law that expresses the threshold value according to sig0 and lew is then established using these datasets. In both cases, the freeboard outputs from CryoSat-2, using TFMRA, is considered as a ground truth. From this reference, a corrective law on the threshold -or the freeboard- outputs from the more sensitive LRM waveforms of Envisat, still using

TFMRA, is deduced. This emphasis the fact that, the heuristic retrackers, which are still necessary to study surfaces too complex for physical models, are approximations which need to be calibrated and validated against representative ground truth.

In summary, this paper presents two novel technics to improve the consistency between Envisat and CryoSat-2 concerning respectively the lead/floe classification and the retracking. The announced results seem very good, in particular in Arctic, but the corresponding plots are not shown (eg, plot showing the R2adj of 0.94 for the freeboards in Arctic) and some comparisons with in-situ data would strengthen the work.

Specific comments

- The second step of the classification is qualified as "supervised" but for me this means supervised by an operator or guided by some external data. It does not seem to be the case here, so could you precise what you mean by "supervised" and "supervised training"?

- The initial classification being done on selected surfaces, above 70°N and avoiding marginal zones, could not this explains the later-on difficulties for these zones?

- There are no quantitative results of the progresses regarding the classification.

- It looks like that you interpolate the heights of floes and leads - and thus the freeboard-all along the track, independently from the surface classification or the distance to the nearest lead. Could you confirm this (defendable) strategy?

- The impressive correlation obtains between Envisat and CryoSat-2 freeboards should be illustrated in order to make the fitting more demonstrative (or at least providing some other statistical characteristics).

- Some comments on the relative importance of the 3 considered parameters (pp, sig0, lew) for the classification and the range correction for Envisat floes would be appreciated.
- Some statements need to be argued (see the Technical Corrections part).

- Some references should be added for: the product that discriminate FYI and MYI, for OSISAF and for DTU15 (even if well known, it is nice to reference them).

- Because of the distinction between sea-ice and leads all along the study, the expression "sea-ice backscatter" is ambiguous as most of the cases it refers to the "surface backscatter" (ie, a mix of sea-ice and leads). This expression can also be simply replaced by "backscatter" as it is a parameter that characterizes the waveform, like the pp or the lew.

- I recommend using the same color-bar theme for the map plots when the purpose is to compare some parameters (of course not necessarily with the same extreme values which depend on the units).

Technical Corrections

- §1, p.2, l.5: what do you mean by "quasi-nadir" (off-nadir data do not measure the right range) and by "run-time measurements" (data are processed off-line).

- §1, p.2, l.6: I dont agree with the sentence: "so accurate that one can see the difference in elevation of the snow surface or the sea-ice surface relative to the sea surface on the leads". All the along-track plots of the ranges show terribly noisy measurements, which justify all the studies to classify the surfaces and filter the ranges. At least you should illustrate or quantify this affirmation.

- §2.1.2, p.3, l.27-28: references for OSISAF and DTU15

- §2.1.2, p.3, l.28: you mean "data filtering"? I suppose you don't analyse the waveforms.

- §2.1.2, p.4, l.2-4: for me the discrepancies between W99 and the snow depth on FYI is mainly coming from the more and more late development of the new sea-ice in the season due to the global warning that strongly impact the Arctic. This delay limits the

possible accumulation of snow on sea-ice. But this worth to be checked.

- §2.1.2, p.4, l.6-17: could you provide with the name of the used product and if possible a reference?

- §2.2, p.4, l.29-32: could you precise the percentage of removed data ?

- §2.3.1,p.5, l.4-6: could you precise what makes you tell that the sea-surface height products are not reliable? Which products?

- §2.3.1,p.5, l.10: I would say more precisely that off-nadir measurements provide wrong ranges.

- §2.3.2,p.6, l.9: with "three classifiers", you mean "three (classifier) parameters"?

- §2.3.2,p.6, l.10: what is the limit for the southern ocean?

- §2.3.3,p.8, l.2 and 7: could we state that 1 « 3? What are the possible impacts?

- §2.4,p.9, l.13: please precise the smoothing function that is used.

- §2.4,p.9, l.25: in what way a 50% threshold for leads and floes is "consistent". Why is it more consistent for CryoSat-2 than for other altimeters?

- §2.4,p.9, l.30: "However" can be removed as the same conclusion is drawn in Guerreiro et al 2017

- §2.4,p.10,F.2: use a unique color-bar

- §2.4,p.10,l.9-10: is there any reason to prefer sig0 than PP ? It could be nice to have also a plot with PP. Visually, the matching with lew is impressive.

- §2.4,p.10,l.5: the sentence here could let imagine that only one monthly value is used in Guerreiro et al. 2017 to establish the correlation. Perhaps you could remove the 2 words "monthly" or precise that all the monthly cells are used.

- §2.4,p.12,l.18: Could you also provide R2 which is more frequently used and for which

we have more references. A plot showing the distribution and the fitting curve would be very welcome. The correlation is just one characterization, among many others, of the fitting and it is not very intuitive.

- §2.4,p.12,l.19: could you display the central Arctic region on one of your maps?

- §2.4,p.12,l.29: could you show on a map the regions where the sig0 and the lew are less correlated? In particular for the lew it is not so obvious.

- §3.1,p.14,l.12-13 and p.15: Could you provide with some quantitative values to illustrate the progress regarding SICCI-1?

- §3.2,p.16,l.23-25: it is not clear whereas all the numbers are related to the current study or some of them concern SICCI-1. For instance the "three cm" line 23 seem in contradiction with the "2.2cm" line 25. Could you provide some quantitative comparison with SICCI-1?

- §3.2,p.19,l.10: typo "Shown are the same months".

- §3.2,p.19,l.21: I don't understand the sentence: "In Antarctic, while the differences are lowered, the overall differences remain larger".

- §4,p.22,l.20-21: and how far are you confident in the AMSRx solution in Antarctic?

---

## Author Comment (AC1) · 15 May 2018

We would like to thank the referee Tom Armitage for his timely review as well as for his the time and effort he spent on it. In the following, we would like to respond to all comments made by the referee and our reasoning behind all our corresponding changes.

General Comments/Suggestions:

**1) Sections 2.3.2 and 2.3.3: I am not familiar with these statistical techniques, and I don't think the wider polar altimetry community will be either, unless there have been other publications on these techniques within the context of polar altimetry? There is a lot of jargon (specialist terms e.g., "k-means", "clustering", "decision trees", "trained forests", etc.) and sentences that are nonsense without specialist knowledge, for example, how does a "decision tree" in a "trained forest" cast a "vote"?! I am imagining woodland creatures and some kind of election! While I appreciate that I could follow the references you have provided to textbooks etc, I think the manuscript would be greatly improved if you could provide a more intuitive explanation of these procedures, as well as some relevant equations and figures if applicable, including specific examples of how this applies to altimeter waveform classification. In particular, I think this is important because this seems to be one of the major developments in the paper, so I think readers should be able to gain an intuitive picture of what is happening in your processing in order that they might develop/reproduce what you have done. The schematic in Figure 1 is sort of useful but there is still a lot of jargon, and it's not clear to me what is happening at each stage.**

We agree with reviewer that this is an essential part of the manuscript and of course, we would like to have this section as intuitive for potential readers as possible. We agree that relatively new, and complex statistical approaches such as the use random forests could most likely be explained in even more detail as it already is (We think one has to appreciate the effort we already put into this instead of a simple mention + reference). However, an additional thoroughly explanation of widely and commonly accepted and proven techniques such as a k-means clustering or simple decision trees might be out of the scope of this manuscript, as it is not intended to give a broad statistical methods review. For the further interested reader we provided all necessary references.

Nevertheless, we made several changes in the mentioned sections and added additional information as well as corrected the mentioned rather figuratively paragraph about the giant democratic group trees.

In 2.3.2:

*"Next, a subset of 1% is sampled at random without replacement (i.e., each original waveform with corresponding surface backscatter, pulse peakiness, and leading-edge width can only appear once) for each month in the MOP and for each sensor independently. This data sample is then separated into three clusters using unsupervised methodology named k-means clustering (MacQueen, 1967; Hartigan and Wong, 1979). This unsupervised method (i.e., without any a-priori information about the data) is widely used to separate input data of $N$ observations into $K$ clusters of equal variance. In our case, based on the input classifier parameters of surface backscatter, pulse peakiness, and leading-edge width, whereby the within-cluster sum-of-squares are iteratively minimized (MacQueen, 1967; Hartigan and Wong, 1979). The result is a 'labeled' data set where each input waveform with corresponding surface backscatter, pulse peakiness, and leading-edge width is labeled as an either sea-ice-type, lead-type, or ambiguous-type waveform.*

*Generally, the preselection of the number of clusters can be a problem when utilizing k-means clustering. However, while we tested a higher number of initial clusters with perspective of later reunion of similar clusters, a separation into just three clusters turned out to be sufficient. Overall, lead waveforms account for a smaller fraction of the total measurements than sea-ice waveforms. Because of this and the fact that k-means clustering tends towards generating equal-size clusters (this is a presumption of k-means clustering algorithms), sole use of k-means clustering for the complete data set was not feasible."*

In 2.3.3:

*"Random forests are based on multiple decision trees. A decision tree is a rather simple statistical tool to predict data categories based thresholds. Over several steps, the input data set is split at each step (called a 'node') based on a threshold of a given parameter until all input data is categorized. When visualized, a decision tree resembles a tree with an increasing numbers of branches, leading to the final categories (Breiman, 2001)."*

As well as some smaller changes to the whole sub-section to increase clarity about the random-forest procedure.

**2) How are your results affected by just using a fixed retracking threshold for Envisat lead waveforms, rather than including lead and floe waveforms in the tuning/fitting procedure? The reasoning for retracking near (or at) the maximum power for lead waveforms is equally valid for CS-2 and Envisat, i.e., specular scattering from leads reduces the effective footprint to the size of the lead, which in turn gives you a return which is close to the transmitted pulse (convolution with a delta function rather than the flat-surface impulse response). Using a single retracker for all CS2 waveforms and essentiallytwo separate retrackers for Envisat leads and floes represents an inconsistency with your approach that I don't feel is justified.**

While physically not correct, the rather empirical solution to use a 50% threshold for the retracking of leads and sea ice for CryoSat-2 using the TFMRA retracker results in the overall best and most plausible results (also compared to validation data, e.g., from EM measurements). While we agree that this is not in any case physically justified, changing our 'reference' without a proper additional validation does not seem justified either.

As the reviewer explains, retracking near the maximum power is the most logical choice for leads, which is also, why we did not aim for an adaptive procedure for Envisat. Additionally, as seen in the paper from Guerreiro et al (2017), using a lead threshold of 50% for leads from Envisat results in unrealistically negative freeboard estimates.

Moreover, using different thresholds for the same retracker algorithm is in our understanding something different than using two completely different retracker algorithms as it was done during SICCI-1 and is a definite improvement by providing a consistent retracking methodology.

**3) Related to this, I appreciate that your fitting procedure is essentially try to match the Envisat freeboard to the CS2 freeboard by tuning the Envisat retracking threshold. But couldn't you simply skip a stage here and fit the Envisat waveform parameters (LEW/sigma-0) to the CS2 freeboard directly?**

While we appreciate the reviewers' suggestion, we do not agree that the proposed procedure would feature the same intuition. Fitting waveform parameter directly to the freeboard would ignore the fact

that sea ice with the same surface features can have different freeboards. Retracking preserves the time delay information, which is not included in the shape parameters but in the position of the waveform in the range window.

**4) Section 3.1: You get a better match with the surface type classification than in SICCI-1, but isn't this by construction? Haven't you tried to match the number of waveforms classified as leads and floes for the two satellites, or have I misunderstood something? Further, I wouldn't necessarily expect there to be agreement between the number of leads/floes detected by the two instruments, simply because of the different footprints – from physical/geometric arguments I would expect far more 'ambiguous' waveforms in the Envisat data. More encouraging is the broad spatial agreement between the lead/floe distributions.**

As mentioned by the reviewer, we indeed see much more ambiguous-type waveforms in the Envisat data and that especially in the Antarctic. However, we also achieve a much better overall spatial agreement of occurrences. While we of course intended to mirror CryoSat-2 patterns of lead-/and sea-ice-occurrences with Envisat, the achieved results are not 'constructed' in a comparable way, as the adaptive retracker threshold procedure. As it is mentioned in the manuscript, all classification is always done for each sensor separately, i.e., while the used procedure is consistent, the methodology is applied for each sensor independently.

**5) Section 3.2: Similarly, isn't the small observed difference in freeboard by construction? Here, I think the manuscript would be greatly improved by comparison of the two satellites with independent radar freeboard measurements, e.g., by combining the IceBridge laser and snow radar.**

Here, the reviewer is correct, as this agreement is indeed through the applied tuning mechanisms. However, no procedure is perfect which is why of course we need to stress the overall very good agreement that our methodology can achieve (but also its limitations e.g. in the Antarctic or in the inter-seasonal variability). The procedure still relies on the different waveform parameters to decide on the best threshold to retrack the freeboard height and there are areas and times where it works better or worse (which we also highlight in the manuscript).

Specific Comments:

**The title is clunky, I would suggest "Consistent retrievals of Arctic and Antarctic sea ice freeboard from Envisat and CryoSat-2"**

We would like to thank the reviewer for the hint. While we did not follow the exact suggestion, we agree that the title might not have been put together elegantly. Furthermore, from the comments we received from all of the three reviewers, we came to the decision that the title was not chosen specifically enough for the purpose of this manuscript. We changed the title to read:

*"Empirical Parametrization of Envisat Freeboard Retrieval of Arctic and Antarctic Sea Ice Based on CryoSat-2: Progress in the ESA Climate Change Initiative"*

**Page 1, Line 2: I suggest "…estimation over recent years, however, precursor…"**

We would like to thank the reviewer for his suggestion and changed that accordingly.

**Page 1, Line 13-15: "cover" should be "extent". Also, join the sentences "…Meier et al), while Antarctic sea ice extent…"**

We changed that.

**Page 1, Line 15-16: I suggest the following "…(Turner et al). Arctic sea ice is also thinning, as observed by…"**

We changed that.

**Page 1, Line 24: I suggest "…that measurement of sea ice thickness at circumpolar scales in both polar regions…"**

We changed that.

**Page 2, Line 4: really this type of processing dates back to at least Laxon (1994), "Sea ice processing scheme at the EODC".**

We would like to thank the reviewer for pointing this out to us. We added the provided reference.

**Page 2, Line 5: I'm not really familiar with the term "run-time" within the context of altimetry, could you explain or use a more familiar term.**

In accordance with a similar remark from reviewer 2/3, we changed that sentence to read:

*"In a first step, the echo power waveforms are classified as returns from either sea-ice floes or returns from the sea surface of leads between sea-ice floes."*

**Page 2, Line 6-7: "so accurate that…", this isn't really the case for individual lead/sea ice measurements due to speckle noise. Also, explain explicitly that this elevation difference is termed the freeboard, otherwise the next sentence might not make sense to people unfamiliar with this term.**

Reviewer 2/3 also questioned this part so we decided to change it to read:

*"These measurements are then converted into distance measurements that let one calculate the elevation difference of the snow surface or the sea-ice surface relative to the sea surface in the leads."*

**Page 2, Line 9-11: "not true" – I'm aware of all the studies on this (including some of my own work!), but I would still argue that such a strong statement on this issue is still not possible. I would suggest "When estimating sea ice thickness from radar altimeters, it is often assumed that…", and you should provide a more extensive list of studies that might suggest otherwise.**

We rephrased this part slightly following the reviewers suggestion:

*"When estimating sea-ice freeboard from radar altimeters, it is often assumed that the retrieved distance over sea ice using Ku-band radar always coincides with the snow/ice interface. However, this*

*assumption is not true, especially for a highly stratified sea-ice snow cover and/or for multi-year sea-ice regimes."*

However, we still think the statement made is valid as a formulation with "not always" clearly implies that there are of course cases where the returned distance actually coincides with the snow/ice interface. But that is clearly not always the case.

**Page 2, Line 18: I believe the Envisat altimeter was Radar Altimeter 2 (RA-2)?**

This is correct. We changed that.

**Page 4, Line 32: Is this filtering important? How many waveforms are removed?**

This filtering step remains from the processing done in during SICCI-1. While the number of rejected data values is potentially small, flag names suggest that it is better to have them removed nonetheless.

**Page 4, Line 33-Page 5, Line 1: Do you also apply a land mask filter? Are the inbuilt land surface type flags accurate enough to catch all land contaminated waveforms?**

We currently rely on the built-in surface-type flags in the CryoSat-2 as well as the Envisat product and use as stated in the manuscript all waveforms flagged as 'Ocean'. No further masking is applied in that matter.

**Page 5, Line 4-8: I think what you are saying is that distinguishing leads is essential in order to estimate the instantaneous sea level anomaly along track?**

In accordance with a comment from reviewer 3 we changed that paragraph to read:

*"The surface-type classification is a crucial part in the processing chain, because the detection of leads is essential for determining the instantaneous sea-surface height anomaly with respect to the mean sea-surface height at the ice-floe location. The resulting sea-surface height at the ice-floe location in turn is used as the reference from which the sea-ice freeboard is calculated."*

**Page 5, Line 9-14: Before this paragraph you should explain why it is possible to distinguish leads and floes to begin with i.e., explain the different surface scattering characteristics. Otherwise, this paragraph is not clear.**

We agree with the reviewer and changed that first part to read:

*"In general, leads feature a specular reflection due to their rather smooth surface, whereas sea ice features a diffuse reflection due to a higher surface roughness. With smaller instrument footprint sizes, less surface-type mixing occurs and the return signal is easier to classify. However, leads often dominate acquired waveforms due to their specular reflection. Off-nadir leads still represent sources of strong backscatter and therefore result in false range estimates."*

**Page 5, Line 12: "footprint of 2km", this is the pulse-limited footprint. "increase up to 10km (Chelton et al.)", I don't think Chelton was talking about off-nadir ranging to leads, he was talking about the effect of significant wave height on the pulse limited footprint, which is fundamentally different i.e., strong off-nadir backscatter in the case of leads vs. large surface roughness in the case of high SWH.**

We removed the latter part.

**Page 5, line 18: "sea ice backscatter", you mean $\sigma^0$?**

Yes. Based on a comment of reviewer 3, we changed 'sea-ice backscatter' into 'surface backscatter' throughout the manuscript to avoid confusion as the parameter is also used to differentiate between leads and sea ice.

**Page 5, Line 29: young, thin ice areas, cause specular reflections, you should add a citation.**

We added the following reference:

*Zygmuntowska, M., Khvorostovsky, K., Helm, V., and Sandven, S.: Waveform classification of airborne synthetic aperture radar altimeter over Arctic sea ice, The Cryosphere, 7, 1315-1324, https://doi.org/10.5194/tc-7-1315-2013, 2013.*

**Page 5, Line 10-31: surely the rejection rate could be decreased as well?**

It surely can, this is in our cased achieved by the proposed way of using monthly thresholds. Estimating thresholds based on a single month however leads to misclassifications/rejections in other months, whereas using all data together potentially also results in rather soft thresholds. This however might allow rather ambiguous signals to be taken into the freeboard retrieval.

**Page 9, Line 24-Page 10, Line 3: This is all rather unclear to me.**

In response to some suggestions made by reviewers 2 and 3 we made changes to this paragraph. This should be clearer now.

**Page 10, Line 1; Figures 2-5: Show the pulse peakiness maps as well, also show the multi-year ice mask for comparison.**

We added the pulse peakiness as well as the resulting Envisat freeboard estimates after application of our adaptive retracker threshold procedure to the Figures. In panel a), showing the freeboard difference between Envisat and Cryosat, we also added the 50% MYI fraction threshold line.

**Page 11, Line 2: Why disregard PP in the fitting procedure? Presumably you tried different iterations but discarding PP gave you the best result?**

We chose sig0 over pp for its more direct physical relation to surface roughness. However, both measures are correlated quite well. Nevertheless, there are differences between pp and sig0 as shown in the new Figures 2-5. In order to keep things as simple as possible and in order to rely on as few

parameters as possible, we decided to use the leading edge width and sig0. As described in the manuscript, the resulting fits are very good based on the given measures.

**Page 12, Line 3-11: The fitting procedure is not clear to me at all. What are you fitting to what? What is the "x-y" plane? Perhaps a diagram/Figure could help to illustrate this?**

We agree and rephrased the first paragraph to read:

*"In the next step, we derive a functional relationship between optimal-threshold values and the waveform parameters of surface backscatter/leading-edge width for our adaptive threshold range-retracking. Therefore, we first average all optimal-threshold values during the mission-overlap period (MOP) for bins of 0.25 dB for the surface backscatter and 0.025 for the leading-edge width, respectively. Here, we use a three dimensional coordinate system with average optimal threshold (z-axis) against leading-edge width (x-axis) and surface backscatter (y-axis)."*

We add here the following sketch for further illustration, where the x-axis represents the leading-edge width, y-axis the surface backscatter and z-axis the optimal Envisat retracker threshold.

[Figure]

**Figure 10: Could you change the x-axis scale to ~ -25cm to ~ 40cm to make the figure clearer**

We changed that for Figures 10 and 11.

**Page 16, Line 26-34: Could the difference with Guerreiro be explained by the speed of light propagation correction, or do you both apply it in the same way?**

In contrast to Guerreiro et al. who use a correction proposed by Kwok & Cunningham 2015 depending on snow depth and density, we apply a speed of light reduction in the snow pack by using a fixed factor of 0.22. However, the resulting difference should be very small and not be able to explain the differences on its own.

---

## Author Comment (AC2) · 15 May 2018

We would like to thank referee Nathan Kurtz for his valuable comments and suggestions. We appreciate the feedback by the referee and would like to go through it point by point and highlight our changes accordingly.

General Comments/Suggestion:

**In particular I think there are flaws in the CryoSat-2 retracking procedure which need to be addressed prior to publication, these are noted below. Particularly I note the need to better refine the lead tracking procedure and verify this through direct elevation comparisons between the measurements. Independent validation data of the CryoSat-2 data set through comparison with field campaigns such as CryoVEx or IceBridge are also needed. Additionally, only freeboard differences are plotted so it is difficult to evaluate whether the retrieved freeboards themselves are accurate. Some maps and statistics of the actual retrieved freeboard are needed. This is especially important for the Antarctic region where due to the complexity of the surface prior studies with satellite radar altimeters have not demonstrated the capacity for obtaining accurate measurements.**

We appreciate the reviewers suggestions and comments. Especially in the case of not shown actual freeboard results, the manuscript had a clear lack of information. We added the resulting Envisat freeboard as well as the gridded Envisat pulse peakiness (suggested by another reviewer) to Figures 2-5.

Referring to the reviewers concern about the lead retracking in the Cryosat product, we would like to mention previous studies that compared the Cryosat product to in-situ data as well as other products (e.g. UCL/CPOM):

Ricker, R., Hendricks, S., Kaleschke, L., Tian-Kunze, X., King, J., and Haas, C.: A weekly Arctic sea-ice thickness data record from merged CryoSat-2 and SMOS satellite data, The Cryosphere, 11, 1607-1623, https://doi.org/10.5194/tc-11-1607-2017, 2017.

Haas, C., Beckers, J., King, J., Silis, A., Stroeve, J., Wilkinson, J., Notenboom, B., Schweiger, A., & Hendricks, S. (2017). Ice and snow thickness variability and change in the high Arctic Ocean observed by in situ measurements. Geophysical Research Letters, 44, 10,462–10,469. https://doi.org/10.1002/2017GL075434

Ricker, R., Hendricks, S., Helm, V., Skourup, H., and Davidson, M.: Sensitivity of CryoSat-2 Arctic sea-ice freeboard and thickness on radar-waveform interpretation, The Cryosphere, 8, 1607-1622, https://doi.org/10.5194/tc-8-1607-2014, 2014.

While we agree that a correct Cryosat reference is important for our procedure (also mentioned by the other reviewers), we think the general plausibility of this data set was shown in other studies. In addition, the framework of this work is the ESA Climate Change Initiative, which advices the use of existing algorithms whenever possible. Thus, the focus of this manuscript lies on matching Envisat freeboard retrievals to those of Cryosat-2 based on Envisat waveform characteristics. While changes in the Cryosat freeboard algorithm would impact the resulting Envisat freeboards, we expect the presented procedure to achieve a comparable fit between the two radar altimeter generations.

We rephrased parts of the Introduction to further stress this approach of using existing algorithms:

*"In this study, we focus on deriving an inter-mission consistent waveform interpretation scheme over sea-ice areas for Envisat and CryoSat-2 in the framework of the second phase of SICCI (SICCI-2).*

*Therefore, the focus of this study lies not in a further optimization of the CryoSat-2 freeboard retrieval, but in the application of an evaluated methodology as is (Ricker et al.; 2014). Based on this approach, we want to find an optimal way to match the freeboard retrieval of Envisat to that of CryoSat-2 and build a consistent sea-ice freeboard data record that takes the different sensor configurations and differing footprints between both sensors into account.*"

We do also acknowledge the limitations of the current empirical CryoSat-2 freeboard retrieval, but the development and validation of an algorithm evolution for SAR sea ice altimetry is beyond the scope of this study. To highlight the ESA CCI approach on relying on existing algorithms, using CryoSat-2 as a reference for Envisat, we changed the title of the manuscript to read:

*"Empirical Parametrization of Envisat Freeboard Retrieval of Arctic and Antarctic Sea Ice Based on CryoSat-2: Progress in the ESA Climate Change Initiative"*

We of course replied to all suggestions made by the reviewer on this topic in the corresponding specific comments below.

Specific Comments:

**P2, second paragraph: The wording choice is a bit awkward in parts...I'm not sure what quasi-nadir run-time measurement means here. "...which are so accurate" could be rephrased better.**

The other reviewers also highlighted this paragraph as rather unclear so we changed it to read:

*"In a first step, the echo power waveforms are classified as returns from either sea-ice floes or returns from the sea surface of leads between sea-ice floes. These measurements are then converted into distance measurements that let one calculate the elevation difference of the snow surface or the sea-ice surface relative to the sea surface in the leads. Here, one can differentiate between the height difference between the top of the snow surface and the sea surface (i.e., the total freeboard) and the height difference between the sea-ice surface and the sea surface (i.e., the sea-ice freeboard)."*

**P2 L18: "a the"**

We changed that.

**P4: If you have daily passive microwave measurements for snow depth retrievals then why is a climatology used for the Antarctic?**

It has been shown in a number of publications that the snow depth based on passive microwave data can be substantially biased due to various physical properties of the sea ice and the snow itself, making the retrieved snow depth noisy and unrealiable at times. Using a climatology suppresses this noise. As the focus of this manuscript is on the possibility to match Envisat freeboard retrievals to those of CS-2 ones based on Envisat waveform characteristics (see last paragraph on page 1) we find it justified if not even mandatory to use a consistent snow depth on sea ice data set. We are aware of the fact that using a climatology is not ideal when it comes to the derivation and geophysical interpretation of a sea-ice thickness time series.

**P5 L17-19: The use of SAR processing on CryoSat-2 will impact both the leading edge width as well as peakiness, I wouldn't expect these value to be equivalent to a pulse limited radar system for lead discrimination.**

This is correct and is also not stated this way in the manuscript. While the overall used scheme is consistent, the resulting thresholds are not. From the provided threshold values in the appendix, one can see that the resulting thresholds are indeed different for the two sensors. However, what is meant here is that instead of relying on classifier parameter such as the stacked standard deviation, which would be available for CryoSat-2, but not for Envisat, we only use parameters that are available to both.

**Section 2.3.2: Some further details on the k-means clustering is needed. Were the peakiness, leading edge width, and backscatter used here? What exactly is coming from the three clusters?**

We changed several parts in the paragraph as also Reviewer 1 suggested changes here to increase the paragraphs clarity to the potential reader. To answer the reviewers question: Yes the three classifier parameters are used here and the result is a labeled training data set that can be used as training data for the second step in the surface-type classification procedure.

*"Next, a subset of 1% is sampled at random without replacement (i.e., each original waveform with corresponding surface backscatter, pulse peakiness, and leading-edge width can only appear once) for each month in the MOP and for each sensor independently. This data sample is then separated into three clusters using unsupervised methodology named k-means clustering (MacQueen, 1967; Hartigan and Wong, 1979). This unsupervised method (i.e., without any a-priori information about the data) is widely used to separate input data of $N$ observations into $K$ clusters of equal variance. In our case, based on the input classifier parameters of surface backscatter, pulse peakiness, and leading-edge width, whereby the within-cluster sum-of-squares are iteratively minimized (MacQueen, 1967; Hartigan and Wong, 1979). The result is a 'labeled' data set where each input waveform with corresponding surface backscatter, pulse peakiness, and leading-edge width is labeled as an either sea-ice-type, lead-type, or ambiguous-type waveform."*

**Section 2.3.3: Same here for the need for further details. What is the training data set that was used, and how was this selected? How was it clear that the method separated leads and floes other than the fact that they had expected values for peakiness and backscatter? Was any validation done of the results to assess the quality of the classification?**

In addition to the changes in the aforementioned paragraph, we also added more information to the subsection about the random forest classifier. We hope that we answered the reviewers question about the training data in the section about the k-means clustering (see above). The selection is rather simple statistics (through grouping of equal variance and increasing cluster homogeneity).

We want to argue that the presented results in the manuscript highlight the benefits and the capabilities of the proposed method. We highlight results from both, orbital as well as gridded data that again can be compared to all waveform parameters (Figures 2-5) as well as the resulting freeboard maps and differences. We think added information about sub steps would only lengthen the manuscript without adding much additional information.

However, we wanted to provide the reviewer with an additional Figure of the randomly picked and averaged Arctic Envisat waveforms in this response letter to persuade him from the resulting data quality. However, no additional validation was conducted.

[Figure]

While the average of the ambiguous waveforms does not look too bad, looking ad individual waveforms reveal the very high amount of noise and inherent surface-type mixture. Nonetheless, an improved future surface-type classification approach might be able to enhance the number of classified valid waveforms.

**Section 2.4: The need for different thresholds for sea ice leads and floes from CryoSat-2 was shown in Kurtz et al., 2014. This should apply to Envisat since both operate on the same physical principle: the effective geometrical area of the lead return is very small causing a radar return which is close to the transmit pulse shape. As both satellites have the same bandwidth the transmit pulse shape should be very similar for both CryoSat-2 and Envisat. However, for sea ice floes the pulse-limited footprint size of Envisat should require a different threshold than the unfocused SAR footprint of CryoSat-2. This implies the threshold chosen for CryoSat-2 floe returns needs to be adjusted. No matter the methodology used though, some validation of the choice of thresholds needs to be done and I think that is lacking in the manuscript. Note too that the approach described in this section assumes the threshold used for CryoSat-2 is a control data set to which the Envisat data is tied, this means the threshold selected for the CryoSat-2 data set is of utmost importance. Thus some validation of this to demonstrate it is correct is sorely needed.**

In general, we agree with the reviewer. We followed the described principles in this manuscript. We also agree that in that in a future algorithm evolution, it is our goal to have an adaptive threshold procedure for both, Envisat and Cryosat-2 tuned based on a large amount of different validation data. However, as mentioned above, the goal in the current project was also to use as many as established algorithms and procedures as possible and we likely stretched to the limit with our current implementation.

The reviewer is also correct in his statement about the pivotal role of and importance of the used CryoSat-2 data for the described principle. We agree that the 50%/50% choice of retracker thresholds for leads/sea-ice used in our implementation is rather an empirical than a physical choice. Nonetheless, in all validation exercises we conducted so far (see references on top) results appear to be plausible and robust. However, the general method presented in this manuscript evolves around the potential of linking freeboard differences between Envisat and Cryosat to Envisat waveform parameters through an adaptive choice of sea-ice retracker thresholds. This is independent of the absolute precision of the

CryoSat-2 data and rather relative towards it. In the aforementioned future evolution of the procedure, where we will have an adaptive retracker solution for Cryosat as well, the here presented procedure will be still valid, with just small adjustments in the predictors of the optimal threshold formulae. Because of that, we would argue that and intensive validation exercise is out of scope for this methodology manuscript. Aside from that, the ESA CCI project features a large validation exercise part that will be published in itself and will be freely available to all users.

**P11 L8-14: This test should be done on the retrieved elevations (not just freeboard values) between Envisat and CryoSat-2, particularly for leads. That would more clearly demonstrate whether the differences in the threshold algorithms are properly handled.**

We have deliberately chosen to evaluate only sea ice freeboard since absolute elevations, or biases therein, have no practical impact for the quality of sea ice thickness. In addition, we cannot assume that the elevations are identical due to the different orbits of Envisat and CryoSat. This kind of analysis would be required for sea level estimations, which is not the scope of our work.

**P11 L15: How was the optimal value chosen? Was it that which had the smallest mean difference, RMS difference, or something else?**

The optimal threshold was chosen based on the smallest absolute difference between Envisat and CryoSat-2 using the gridded monthly data. Here, the monthly Envisat freeboard was calculated for each TFMRA retracker threshold between 5% and 95% of the first maximum in steps of 5%. For example, in case a threshold of 50% results in a absolute difference of 0.2cm of a given pixel, whereas the 55%/45% thresholds result in higher differences, then the optimal threshold is set to 50% for that pixel given these values of sigma0 and leading-edge width.

**Figures 10 and 13 seem to not match up visually. In Figure 13 there seems to be a far higher spatial coverage of red, indicating a higher Envisat freeboard whereas the distributions in Figure 10 seem to show only small mean differences and a more symmetric distribution. Some clarification on this is needed.**

While the mean difference is indeed small, especially for November 2011 (~2cm; the left panel in Figure 13), the key to this visually appearing 'inconsistency' lies in the histograms (Figure 10, bottom row, second to the left panel). While the CryoSat-2 freeboards peak at a lower range than the Envisat freeboard estimates, we find these larger amount of red in Figure 13a. However, as these histograms are based on all available corresponding freeboard estimates, also all areas where Envisat underestimates CryoSat-2 are taken into account. Nevertheless, a very large amount of differences is centered on zero, which leads to the overall small mean difference.

---

## Author Comment (AC3) · 15 May 2018

We would like to thank referee Sara Fleury for her review and appreciate her valuable comments and suggestions. Here, we would like to go through the made comments point by point and highlight our changes accordingly.

Specific Comments:

**- The second step of the classification is qualified as "supervised" but for me this means supervised by an operator or guided by some external data. It does not seem to be the case here, so could you precise what you mean by "supervised" and "supervised training"?**

In contrast to an unsupervised classification or clustering what we used in the first step, a supervised classification in general involves some kind of predefined or labeled training data. This does not necessarily mean external data and could also be selected manually by an expert or like in our case through an unsupervised clustering as a pre-processing step. In case of the here-used Random-Forest Classifier as a procedure qualified for the term of "machine learning", the labeled training data from the k-means clustering is used to train the classifier so it is able in a second step to classify previously unknown data into the prescribed classes of 'ambiguous', 'lead', and 'sea ice'.

**- The initial classification being done on selected surfaces, above 70N and avoiding marginal zones, could not this explains the later-on difficulties for these zones?**

That is very likely. However, we decided to do it the way we did because of the much larger noise that stems from the higher degree of surface-type mixing, impact of ocean swell, presence of larger areas of thin and new ice in these areas. Because of these very challenging conditions, a proper fit that also ensures very good quality in the central Arctic would be rather difficult.

**- There are no quantitative results of the progresses regarding the classification.**

The reviewer is correct, however, we rate the presented results as of a higher importance compared stating specifics about cluster centers. We present both comparisons based on orbit as well as gridded data that can again be compared to the monthly gridded waveform parameters and the resulting freeboard maps.

The following Figure presents the average waveforms for all three classes. These are randomly picked from the Arctic for Envisat. However, compared to other Figures in the manuscript, we rate this information as less relevant for the overall methodology explanation.

[Figure]

**- It looks like that you interpolate the heights of floes and leads - and thus the freeboard all along the track, independently from the surface classification or the distance to the nearest lead. Could you confirm this (defendable) strategy?**

We are not sure what exactly the reviewer is referring to. We do not interpolate heights of leads and sea ice independent of the surface-type classification. What we assume the reviewer is referring to is that we reject interpolated sea surface height if the distance to the next lead tie point is greater than 200km.

**- The impressive correlation obtains between Envisat and CryoSat-2 freeboards should be illustrated in order to make the fitting more demonstrative (or at least providing some other statistical characteristics).**

Already several figures are dedicated to illustrate the in general very good agreement between the freeboard estimates from Envisat and CryoSat-2 (the histogram visualizations of all monthly gridded freeboard estimates in Figures 10&11 of mutually covered grid cells, the overall resulting cumulative frequencies of freeboard differences in Figure 12, and the resulting gridded freeboard estimates in Figures 13&14). Therefore, I think there are no further Figures needed and would otherwise just lengthen the manuscript without adding a lot of additional information.

**- Some comments on the relative importance of the 3 considered parameters (pp, sig0, lew) for the classification and the range correction for Envisat floes would be appreciated.**

By means of classification, all parameters are important, however, especially sig0 and pp show high importance scores in the random forest classification. For the retracker threshold estimation, sig0 and lew showed the most promising results through their capability to capture the seasonal cycle the best and focusing better on the characteristics of MYI (higher surface roughness e.g.).

**- Some statements need to be argued (see the Technical Corrections part).**

We would like to thank the reviewer for her suggestions and would like to refer to our detailed feedback on those comments in the Technical Comments section.

**- Some references should be added for: the product that discriminate FYI and MYI, for OSISAF and for DTU15 (even if well known, it is nice to reference them).**

We agree. Please see our changes in the Technical Comments part.

**- Because of the distinction between sea-ice and leads all along the study, the expression "sea-ice backscatter" is ambiguous as most of the cases it refers to the "surface backscatter" (ie, a mix of sea-ice and leads). This expression can also be simply replaced by "backscatter" as it is a parameter that characterizes the waveform, like the pp or the lew.**

We agree with the reviewer that this term is rather misleading as also the backscatter from lead-type waveforms are currently referred to as sea-ice backscatter. We changed sea-ice backscatter to surface backscatter in the complete manuscript.

**- I recommend using the same color-bar theme for the map plots when the purpose is to compare some parameters (of course not necessarily with the same extreme values which depend on the units).**

Due to the suggestion of another reviewer we added the pulse peakiness to the Figure Set as well as the resulting Envisat Freeboard. We now use the same color map for all three waveform parameters.

Technical Corrections:

**- §1, p.2, l.5: what do you mean by "quasi-nadir" (off-nadir data do not measure the right range) and by "run-time measurements" (data are processed off-line).**

The reviewer is right and we removed these misleading terms and changed the sentence to read:

*"In a first step, the echo power waveforms are classified as returns from either sea-ice floes or returns from the sea surface of leads between sea-ice floes."*

**- §1, p.2, l.6: I dont agree with the sentence: "so accurate that one can see the difference in elevation of the snow surface or the sea-ice surface relative to the sea surface on the leads". All the along-track plots of the ranges show terribly noisy measurements, which justify all the studies to classify the surfaces and filter the ranges. At least you should illustrate or quantify this affirmation.**

We change the sentence to read:

*"These measurements are then converted into distance measurements that let one calculate the elevation difference of the snow surface or the sea-ice surface relative to the sea surface in the leads."*

**- §2.1.2, p.3, l.27-28: references for OSISAF and DTU15**

We added the following references in the text:

*"[…] sea-ice concentration data obtained from the Ocean and Sea Ice Satellite Application Facility (OSISAF; ftp://osisaf.met.no/reprocessed/ice/conc/v1p2) as well as the mean sea-surface height product provided by the Danish Technical University (DTU; Anderson et al., 2016; ftp://ftp.spacecenter.dk/pub/DTU15/) in its 2015 version"*

**- §2.1.2, p.3, l.28: you mean "data filtering"? I suppose you don't analyse the waveforms.**

The reviewer is correct; we changed the sentence to read:

*"Sea-ice concentration data is used mainly to discard waveforms based on a minimum required sea-ice concentration threshold of 5%, […]"*

**- §2.1.2, p.4, l.2-4: for me the discrepancies between W99 and the snow depth on FYI is mainly coming from the more and more late development of the new sea-ice in the season due to the global warning that strongly impact the Arctic. This delay limits the possible accumulation of snow on sea-ice. But this worth to be checked.**

We agree with the reviewer. This doubt further encourages us to use only 50% of the Warren values over FYI.

**- §2.1.2, p.4, l.6-17: could you provide with the name of the used product and if possible a reference?**

As mentioned in the manuscript, this product results from the reprocessing of co-author Stefan Kern based on the mentioned data products in order to have a consistent product for the complete combined lifespan of ERS, Envisat and CryoSat-2. So far, there is no specific reference for these data other than a section in the ESA-CCI sea ice ECV phase 2 (SICCI2) ATBD for sea-ice concentration: SICCI-P2-ATBD(SIC), Version 1.0, Sep. 2017.

**- §2.2, p.4, l.29-32: could you precise the percentage of removed data ?**

This filtering step remains from the processing done in during SICCI-1. While the number of rejected data values is potentially small, flag names suggest that it is better to have them removed nonetheless. We do not capture the exact number of waveforms that are removed in this step.

**- §2.3.1,p.5, l.4-6: could you precise what makes you tell that the sea-surface height products are not reliable? Which products?**

In accordance with a comment from reviewer 1 we changed that paragraph to read:

"The surface-type classification is a crucial part in the processing chain, because the detection of leads is essential for determining the instantaneous sea-surface height anomaly with respect to the mean sea-surface height at the ice-floe location. The resulting sea-surface height at the ice-floe location in turn is used as the reference from which the sea-ice freeboard is calculated."

The general principle of estimating freeboard from radar altimetry is already summarized in the introduction and does not contribute any benefit at this point in the manuscript.

**- §2.3.1,p.5, l.10: I would say more precisely that off-nadir measurements provide wrong ranges.**

We changed that.

**- §2.3.2,p.6, l.9: with "three classifiers", you mean "three (classifier) parameters"?**

The reviewer is correct. We made changes at several occasions throughout the manuscript to clarify the term "classifiers" as "classifier parameters".

**- §2.3.2,p.6, l.10: what is the limit for the southern ocean?**

This is explained in line 15 on page 6:

"For the Antarctic the same restrictions apply, but waveforms are geographically limited to an area *south of 65°S to exclude the majority of the marginal-ice zone to reduce the impact of ocean swell*"

**- §2.3.3,p.8, l.2 and 7: could we state that 1 « 3? What are the possible impacts?**

While this suggestion by Breiman was definitely made for larger amounts of input parameters, setting m=1 is the best possible way. While the Random Forest Classifier is capable and powerful enough to deal with very large amounts of input parameters, there is no doubt about its quality using only fewer, but very suitable parameters for the data at hand.

**- §2.4,p.9, l.13: please precise the smoothing function that is used.**

We clarified this bullet point:

*"- Smoothing of the oversampled waveforms with a running-mean window-filter size of 11 (Envisat, CryoSat-2 SAR) or 21 (CryoSat-2 SIN) range bins respectively;"*

**- §2.4,p.9, l.25: in what way a 50% threshold for leads and floes is "consistent". Why is it more consistent for CryoSat-2 than for other altimeters?**

Maybe the word "consistent" was misleading at this point. What it meant is that we follow previous work conducted at AWI and use the same retracker threshold setup (being consistent with that work) that so far showed good results. However, we removed the word at this point in the text.

**- §2.4,p.9, l.30: "However" can be removed as the same conclusion is drawn in Guerreiro et al 2017**

We changed that.

**- §2.4,p.10,F.2: use a unique color-bar**

Please refer to our response in the Specific Comments section.

**- §2.4,p.10,l.9-10: is there any reason to prefer sig0 than PP ? It could be nice to have also a plot with PP. Visually, the matching with lew is impressive.**

As the reviewer mentions, the visual correlation between sig0, pp and lew is quite good, and especially high between sig0 and pp for what we saw (please refer to our updated Figures 2-5). To keep things as simple as possible, which one might argue about in the case of fitting a $3^{rd}$ order polynomial plane, we decided to stick to as few parameters as possible and chose in that case sig0 over pp for its more direct relationship to surface roughness.

**- §2.4,p.10,l.5: the sentence here could let imagine that only one monthly value is used in Guerreiro et al. 2017 to establish the correlation. Perhaps you could remove the 2 words "monthly" or precise that all the monthly cells are used.**

We assume the reviewer is referring to page 11, l5 here instead of page 10 as there is no reference to Guerreiro et al. on page 10. We added "monthly-gridded" to clarify that all cells of a month are used for their estimations.

**- §2.4,p.12,l.18: Could you also provide R2 which is more frequently used and for which we have more references. A plot showing the distribution and the fitting curve would be very welcome. The correlation is just one characterization, among many others, of the fitting and it is not very intuitive.**

Please refer to our answer to your general comment. The here-used adjusted $R^2$ is always lower or equal to the normal $R^2$ by definition. We therefore assume there is no valuable additional information from it.

**- §2.4,p.12,l.19: could you display the central Arctic region on one of your maps?**

The central Arctic region is pretty much what one would expect. However, we agree our text in the manuscript suggests a rather vaguer delimitation. We changed the text in the parenthesis to read:

*"[…] (i.e., we excluded the Canadian Arctic Archipelago and the Hudson Bay, but also extensive fast-ice areas like the Laptev Sea) […]"*

**- §2.4,p.12,l.29: could you show on a map the regions where the sig0 and the lew are less correlated? In particular for the lew it is not so obvious.**

Similarly to the reviewers comment above, we clarified this in the text by adding the following text:

*"[…] as well as patterns in surface backscatter and leading-edge width are less correlated in some areas (e.g., the MIZ but also in the central Weddell Sea; […])"*

**- §3.1,p.14,l.12-13 and p.15: Could you provide with some quantitative values to illustrate the progress regarding SICCI-1?**

Schwegmann et al. (2016) focus on the Antarctic and a quantitative comparison is rather speculative due to the limitation we have through diversified snow stratigraphy and surface flooding on Antarctic sea ice. However, visual comparisons between both studies suggest a substantial improvement. For the Arctic, Kern et al. (2015) provide an exemplary visualization of the Envisat SICCI-1 freeboard for the Arctic in March 2010. While we do not cover this period in time in this manuscript, visual comparisons between both studies again suggest a substantial improvement

Kern, S., Khvorostovsky, K., Skourup, H., Rinne, E., Parsakhoo, Z. S., Djepa, V., Wadhams, P., and Sandven, S.: The impact of snow depth, snow density and ice density on sea ice thickness retrieval from satellite radar altimetry: results from the ESA-CCI Sea Ice ECV Project Round Robin Exercise, The Cryosphere, 9, 37-52, https://doi.org/10.5194/tc-9-37-2015, 2015.

**- §3.2,p.16,l.23-25: it is not clear whereas all the numbers are related to the current study or some of them concern SICCI-1. For instance the "three cm" line 23 seem in contradiction with the "2.2cm" line 25. Could you provide some quantitative comparison with SICCI-1?**

The reference to SICCI-1 is based primarily on the paper of Schwegmann et al. (2016) for the Antarctic and some internal analysis that lead to the improvements made in this study. These general impressions and limitations are summarized in the Introduction. There is so far no citable publication concerning the SICCI-1 results for the Arctic. Concerning the 2.2cm statement, we clarified the last sentence to read:

*"The overall maximum monthly average freeboard differences is 2.2cm"*

**- §3.2,p.19,l.10: typo "Shown are the same months".**

We changed that.

**- §3.2,p.19,l.21: I don't understand the sentence: "In Antarctic, while the differences are lowered, the overall differences remain larger".**

We thank the reviewer for pointing this out to us. We clarified the sentence to read:

*"In the Antarctic, while the freeboard differences between both sensors are lowered through applying the here-presented methodology, the overall resulting differences remain larger than the ones estimated for the Arctic."*

**- §4,p.22,l.20-21: and how far are you confident in the AMSRx solution in Antarctic?**

It has been shown in a number of publications that the snow depth based on passive microwave data can be substantially biased due to various physical properties of the sea ice and the snow itself, making the retrieved snow depth noisy and unrealiable at times. Using a climatology suppresses this noise. As the focus of this manuscript is on the possibility to match Envisat freeboard retrievals to those of CS-2 ones based on Envisat waveform characteristics (see last paragraph on page 1) we find it justified if not even mandatory to use a consistent snow depth on sea ice data set. We are aware of the fact that using a climatology is not ideal when it comes to the derivation and geophysical interpretation of a sea-ice thickness time series.

.

---

## Author Response (AR2)

We would like to thank Editor Jennifer Hutchings as well as the Referees Tom Armitage and Sara Fleury for their further comments and helpful suggestions. In the following, we would like to go through these final remarks point-by-point and highlight all changes made to the manuscript.

**Comments by Editor Jennifer Hutchings**

**Minor comments:**

**page 3 line 19: "used input data" seams less jarring as "input data used". And similarly throughout the manuscript.**

We changed that.

**page 4 line 24 "More information is given in Kern (unpublished manuscript, 2016)." Is this manuscript published yet? Check that it is not a technical manual.**

We changed it to a technical manual that was produced during SICCI-2 as it is very unlikely to be every published as a proper manuscript. Thanks for pointing this out.

**Section 2.2 heading. Should this be moved up to above "As a first step, general filtering is applied"? These lines 1 through 8 seam out of place. Where ever these sentences should reside in the manuscript they do need some clarification as to what they are referring to.**

We moved this subsection upwards and into subsection "2.1.1 Altimetry data" where we think it fits better and its purpose is clear in explaining necessary filtering steps in the radar altimetry data used for this study.

Subsection 2.1.1. now reads:

*"For our study, we use geolocated level 1b (L1b) data for both CryoSat-2 and Envisat. In case of CryoSat-2, we make use of all available SIRAL Baseline-C data acquired in synthetic aperture radar mode (SAR) as well as in the SAR interferometric (SIN) mode. However, the specific interferometric information is not used during the processing. No further filtering based on quality control is conducted.*

*For Envisat, we use version 2.1 of the sensor geophysical data record (SGDR). All data is provided by the ESA. Here, we investigated the measurement-confidence data flags in the SGDR for problematic records. All data with 'Packet Length Error' (Flag 0), invalid OnBoard Data Handling (Flag 1), an Automatic Gain Control fault (Flag 4), a Rx Delay Fault (Flag 5) or an Waveform Fault (Flag 6) raised are removed from processing.*

*In the second step, all data is filtered regionally for both hemisphere by latitudinal boundaries to areas where sea ice is present. Data are only considered if located north of 60°N for the Arctic and south of 50°S for the Antarctic.*

*Finally, all processing for both sensors is limited to waveforms flagged as ocean."*

**Make sure all acryonyms are defined. For example is L1b (line 20, page 6) defined earlier. The general readership of The Cryosphere may not understand how satellite products are labled.**

Thank you for pointing this out again. We went through the manuscript again and also changed some minor type errors.

**page 10 line 27-28: " our Envisat sea-ice-freeboard estimates featured an overall smaller variation and range than CryoSat-2 estimates" replace than with "compared to".**

We changed that.

**page 15 line 5: "This is potentially related ice-snow interface" missing "to" at "This is potentially related to ice-snow interface"**

We changed that.

**page 16, line 20: remove hypen from "lead-". Also in other places.**

We also changed that.

**Comments by Referee Tom Armitage**

**Thanks to the authors for their attempts to address my concerns. With the exception of one minor point (below), I am more or less satisfied with the responses you have provided, and I think the manuscript is in fairly good shape for publication.**

**Both Nathan Kurtz and myself picked up on the point of retracker thresholds and the discrepancy between the retracking applied to the two different instruments. I am not going to repeat the points here, but in short, you decided to push back against our recommendations on using a consistent approach between the two instruments, as well as both of our recommendations to perform an independent evaluation using e.g., airborne data.**

**I understand that your reasoning is simply that further development of the CS2 freeboard wasn't part of this study, and you are taking the CS2 data as it is 'off the shelf'. You also explain that you have plans to perform an evaluation, as well as to preform the 'adaptive threshold' procedure to the CS2 data in future. This is fine, as I think I now understand better that the scope of this study was simply to develop the freeboard cross-calibration and nothing else, however, I would like to see this set out clearly and unambiguously in the manuscript. You have set out your purpose at the end of the introduction, but you could add a section in the discussion, or a paragraph in the summary/outlook, simply describing what you have said in response to our comments i.e., you acknowledge there is an inconsistency in using different retracking etc., but that a) the purpose of this paper is simply just the freeboard cross-calibration algorithm, b) you plan on developing a consistent adaptive retracking procedure for CS2 in the future, and c) that the evaluation you have shown isn't independent and you plan on performing a truly independent evaluation further on in the SICCI-2.**

We would like to thank Tom Armitage again for the effort and time he put into this and acknowledge his final remarks. We added the following statements into our summary/outlook section to clarify our intentions and future aims:

*"While the employed surface-type classification is truly consistent between both sensors, there still is an inconsistency with regard to the range retracking in using fixed thresholds for CryoSat-2 in contrast*

*to an adaptive procedure for Envisat. However, the purpose of this study in the framework of SICCI-2 was to use existing data sets and algorithms wherever possible to create a cross-calibrated freeboard algorithm. In the future, we are planning to develop a truly consistent similar adaptive retracking procedure for CryoSat-2 as well. As we take the CryoSat-2 data in this study as correct, the shown evaluation is not independent. A truly independent evaluation will be conducted in a future study."*

**Comments by Referee Sara Fleury**

**I accept the paper as is because the author has answered to most of my questions or suggestions. I still regret that the are no validation with in-situ data but I can understand that would delay and enlarge significantly the paper.**

Thank you again for the time and effort you put into reviewing our manuscript. We will work on a particular validation paper in the future, as it will also state in the manuscript (as also pointed out by Tom Armitage)

On behalf of all authors,

Stephan Paul

[revised manuscript text omitted]